# SRSR: Enhancing Semantic Accuracy in Real-World Image Super-Resolution with Spatially Re-Focused Text-Conditioning

Chen Chen[1,2*]    Majid Abdolshah[1]    Violetta Shevchenko[1]
Hongdong Li[1,3]    Chang Xu[2]    Pulak Purkait[1]
[1]Amazon    [2]The University of Sydney    [3]Australian National University

## Abstract

Existing diffusion-based super-resolution approaches often exhibit semantic ambiguities due to inaccuracies and incompleteness in their text conditioning, coupled with the inherent tendency for cross-attention to divert towards irrelevant pixels. These limitations can lead to semantic misalignment and hallucinated details in the generated high-resolution outputs. To address these, we propose a novel, plug-and-play *spatially re-focused super-resolution (SRSR)* framework that consists of two core components: first, we introduce Spatially Re-focused Cross-Attention (SRCA), which refines text conditioning at inference time by applying visually-grounded segmentation masks to guide cross-attention. Second, we introduce a Spatially Targeted Classifier-Free Guidance (STCFG) mechanism that selectively bypasses text influences on ungrounded pixels to prevent hallucinations. Extensive experiments on both synthetic and real-world datasets demonstrate that SRSR consistently outperforms seven state-of-the-art baselines in standard fidelity metrics (PSNR and SSIM) across all datasets, and in perceptual quality measures (LPIPS and DISTS) on two real-world benchmarks, underscoring its effectiveness in achieving both high semantic fidelity and perceptual quality in super-resolution.

## 1 Introduction

Image super-resolution (SR) aims to restore a high-resolution (HR) image from a low-resolution (LR) counterpart degraded by blur, noise, and compression artifacts or other distortions. This classic problem has broad real-world applications, ranging from photography [18] and digital surveillance [63, 32] to medical imaging [12, 41, 17] and remote sensing [45]. However, restoring both photorealistic details and semantically coherent content under severe degradations remains an open challenge. Early deep-learning-based approaches often assume simple, known degradations (e.g., bicubic or Gaussian downsampling) and investigate new architectural designs [3, 4, 8, 9, 24, 28, 38, 67]. Subsequently, super-resolution methods based on generative models like GANs [11] and diffusion models [16, 29] can produce realistic details [62, 46, 19, 49], yet they frequently struggle on real-world LR image inputs that deviate from the above assumptions, resulting in clear artifacts in the super-resolved outputs. Consequently, recent research adopts more complex and realistic degradation pipelines (e.g., Real-ESRGAN [46]) and exploits the powerful Stable Diffusion (SD) prior [36], as in StableSR [44] and DiffBIR [25], to improve generalization in real-world ISR. However, these methods overlook textual guidance, thus risking semantic misalignment.

A recent trend is to incorporate text priors to guide super-resolution via a text-to-image Stable Diffusion pipeline, leading to better semantic fidelity [58, 59, 31, 54, 53]. Among these methods, the degradation-aware prompt extractor (DAPE) proposed by SeeSR [54] demonstrates superior

---

*Work done during internship at Amazon.

39th Conference on Neural Information Processing Systems (NeurIPS 2025).

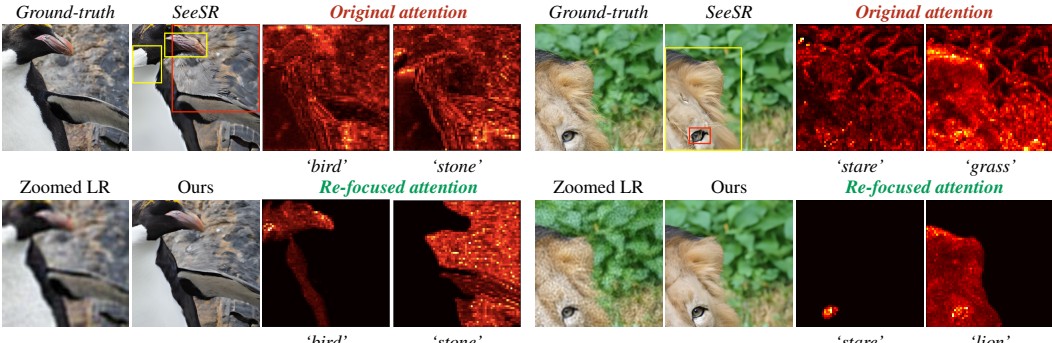

Figure 1: Illustrations of how inherent cross-attention can be misled by irrelevant tokens, resulting in semantically incorrect restorations (top). Left: the baseline mistakenly associates the stone region with the token 'bird', and the animal's neck and beak with 'stone', producing wing-like artifacts on the stone and unnatural textures on the animal. Right: attention for 'stare' is scattered across irrelevant patches, and both the eye and lion face incorrectly respond to 'grass', introducing hallucinated textures. We propose re-focusing cross-attention by constraining the influence of each text token to its grounded region, yielding sharper and semantically aligned reconstructions (bottom).

performance in terms of both degradation-awareness and efficiency compared to other off-the-shelf models used like ResNet [14], YOLO [33], BLIP [23] and LLaVA [26] employed in other works. OSEDiff [53] further trains a one-step model that also leverages DAPE as the prompt extractor.

Despite these advances, text conditioning can induce hallucinations, causing ambiguous semantics in the restored results, as illustrated in Fig. 1. We identify three main limitations in current designs: *Firstly*, existing approaches rely exclusively on cross-attention to inject text prompts, but tokens often leak onto unrelated regions, leading to mismatched guidance and semantic ambiguity. *Secondly*, although DAPE is more degradation-aware than other prompt extractors, it remains only partially robust to severe degradations, making accurate prompt extraction challenging. Providing erroneous text prompts can degrade the fidelity of super-resolved outputs more severely than omitting text guidance altogether. *Thirdly*, prompt extraction methods do not guarantee full-image coverage, leaving ungrounded regions without guidance from relevant semantics and vulnerable to the influence of unrelated text prompts. A recent work [40] exploits Mask2Former [5] trained on ADE20K [70] to achieve full-image coverage by assigning dense segmentation labels to every pixel. However, its segmentation model lacks degradation-awareness and is restricted to 150 categories, limiting its scope for real-world applications that demand open-vocabulary understanding. Bridging these limitations to achieve fully coherent, semantically correct super-resolution remains an open challenge.

To address the aforementioned challenges, we propose a novel framework that conducts *Spatially Re-focused Super-Resolution (SRSR)*. We first apply *visual grounding* to filter out less relevant tags and generate tag–region pairs via segmentation masks. Based on these, we introduce *Spatially Re-focused Cross-Attention (SRCA)*, which constrains each tag's influence to its corresponding spatial region, effectively mitigating semantic hallucinations from the first two limitations. To further improve restoration in ungrounded regions, we introduce a *Spatially Targeted Classifier-Free Guidance (STCFG)* mechanism that selectively disables classifier-free guidance in these regions to avoid undesired text-conditioning effects. Notably, our method operates exclusively at inference time, eliminating the need for additional training or fine-tuning. As a result, it serves as a lightweight, plug-and-play module compatible with any cross-attention-based SR approach that utilizes text priors.

Our results show that incorporating SRSR into both the 1-step OSEDiff and the 50-step SeeSR baselines yields significant improvements in full-reference metrics across synthetic and real-world datasets. When compared to seven other state-of-the-art baselines, our method consistently outperforms all alternatives in fidelity measures PSNR and SSIM on every dataset, and achieves the top performance for perceptual metrics LPIPS and DISTS on the two real-world datasets (RealSR and DrealSR) while ranking second on the synthetic DIV2K set. In summary, our contributions are threefold: *First*, we highlight underexplored gaps in existing methods that impede semantically accurate super-resolution. *Second*, we propose a novel SRSR framework that effectively alleviates semantic ambiguity through spatially re-focused cross-attention (SRCA) and spatially targeted CFG (STCFG), while remaining a lightweight, inference-time plug-and-play module for any cross-attention-based SR approach with text priors. *Third*, SRSR achieves new state-of-the-art performance against seven strong baselines.

## 2 Related work

### 2.1 Non-diffusion-based super-resolution methods

Starting with CNN-based approaches, SR has been extensively explored through models that directly map LR images to HR outputs in an end-to-end manner [21, 69, 13, 9, 39, 6]. Many CNN-based methods focus on designing deeper or wider networks and improving attention mechanisms to enhance SR performance [48]. With the rise of generative models, GANs have gained popularity by using adversarial frameworks where the generator creates HR images closely matching real image distributions, thereby improving perceptual quality and realism. Real-ESRGAN [46] advanced this by introducing a high-order degradation model and modifying the U-Net discriminator to enhance stability and visual performance in real-world SR scenarios. Building on this, uncertainty and semantic-aware approaches have been proposed [27, 42, 22, 61, 30]. Uncertainty-Aware GANs [27] integrated pixel-level uncertainty into adversarial training for fine-grained feedback, while SeD [22] incorporated semantic information into the discriminator to guide the generator in producing semantically aligned textures. CAL-GAN [61], another context-aware method, employs a mixture of classifiers to handle image patches based on content, improving discriminator capacity for photo-realistic SR. In parallel, SFT-GAN [47] demonstrates how spatially-varying categorical priors derived from semantic segmentation maps can be integrated into SR networks through Spatial Feature Transform (SFT) layers, enabling region-specific modulation of features and the recovery of textures that are more realistic and faithful to underlying semantics. However, these methods primarily focus on broader image-level semantic context rather than explicitly leveraging object-level understanding within images for SR.

### 2.2 Diffusion-based super-resolution methods

DDRM and DDNM [19, 49] were the earlier efforts that develop DDPM-based SR methods. Subsequently, Stable-Diffusion-based SR methods like StableSR [44] and DiffBIR [25] were proposed to tackle the more challenging real-world image super-resolution (Real-ISR) task by leveraging Stable Diffusion (SD)'s strong prior that trained using billions of image-text pairs. A recent trend is to incorporate text priors to guide generation in text-to-image SD, leading to better semantic fidelity. For example, CoSeR [37] highlights the potential of bridging image and language understanding using diffusion-based priors and cognitive embeddings. Also, PASD [58] extracts text prompts via off-the-shelf models (ResNet [14], YOLO [33] and BLIP [23]) and leverages ControlNet [65] without modifying SD's pre-trained weights. SUPIR [59] and XPSR [31] employ LLaVA [26], while SeeSR [54] proposes a degradation-aware prompt extractor (DAPE) that improves on the Recognize Anything Model (RAM) [68] for LR images. OSEDiff [53] further trains a one-step model that also leverages DAPE as the prompt extractor, demonstrating its superiority over larger vision-language models like LLaVA. Most recent works like SegSR [55] and HolisDiP [40] propose to additionally use semantic segmentation models to extract prompts with more comprehensive coverage. Despite these advances, ensuring the accuracy of the extracted prompt and the robustness of its cross-attention injection remains difficult under heavy degradation, as incorrect text guidance can still trigger hallucinations and semantic ambiguity in the restored outputs.

## 3 Method

### 3.1 Research gaps

Recent work has shown that super-resolution (SR) methods based on Stable Diffusion (SD) hold promise, primarily due to SD's powerful prior learned from billions of image-text pairs. However, when operating on degraded LR images, semantic errors can arise, leading to inaccurate reconstructions. Recent studies [58, 54, 59, 53] have demonstrated that leveraging SD's text-conditioning, guided by text priors extracted from LR images, can yield more semantically-aware SR results. Nevertheless, hallucination is often observed due to three major limitations of the current designs:

**Diverted cross-attention**. Existing approaches rely on cross-attention to implicitly associate text prompts with image pixels. However, we observe that these associations are often misaligned, with text tokens attending to irrelevant regions. As shown in Fig. 1 (top row), the baseline cross-attention maps frequently misattribute tokens such as 'bird', 'stone', 'stare', and 'grass' to unrelated image areas, leading to semantic ambiguity and perceptual artifacts. The output's semantic fidelity is further degraded when the extracted text contains incorrect or loosely related tags. For instance, in Fig. 2 (top row), 'tower' is loosely relevant to 'building', but the latter is mainly used for guiding the restoration of the building regions. In contrast, 'tower' pays significant attention to regions highlighted by the yellow bounding boxes, which actually correspond to tree-like structures, introducing hallucinated

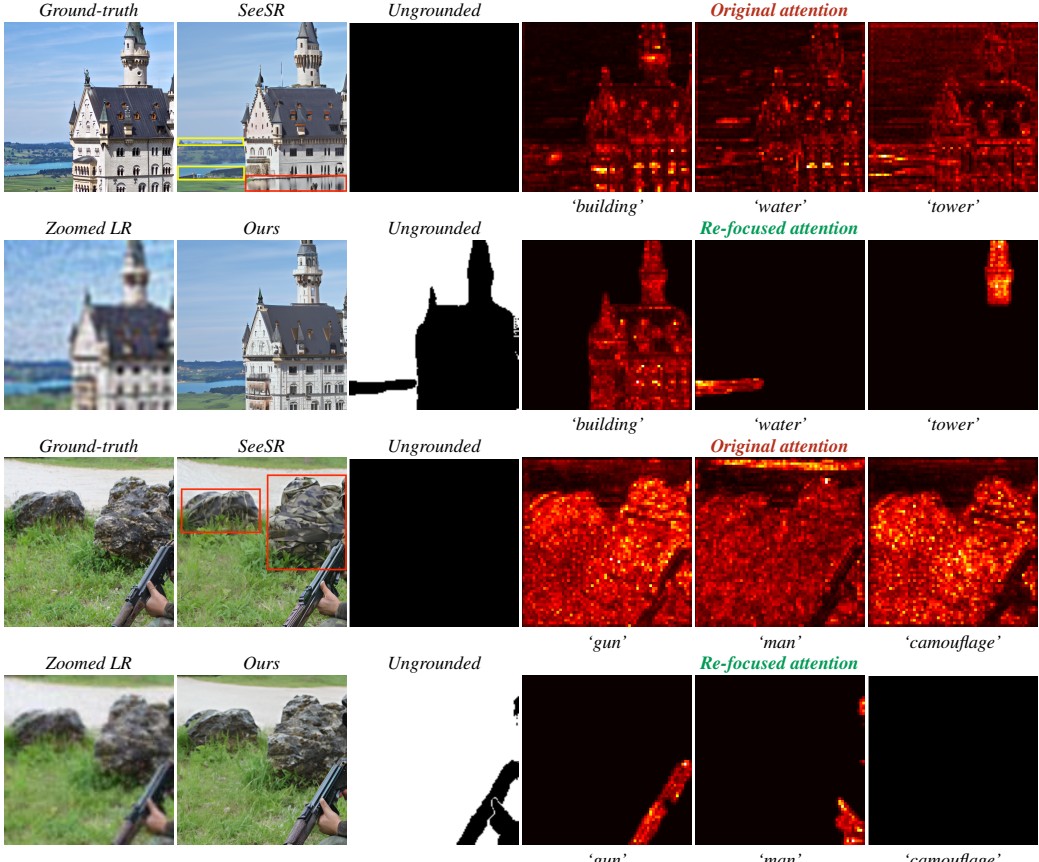

Figure 2: Analysis of how our proposed *Spatially Re-focused Cross-Attention (SRCA)* and *Spatially Targeted Classifier-Free Guidance (STCFG)* improve semantic fidelity in text-conditioned super-resolution. The *Ungrounded* mask highlights regions where no textual tag can be confidently grounded. Existing methods rely solely on inherent cross-attention, allowing global or irrelevant tokens to influence all regions. *SRCA* addresses this by limiting each token's influence to its corresponding grounded region only, reducing semantic confusion. However, this leaves ungrounded regions only associated with global tokens (e.g., EOS, padding, punctuation), where summary tokens like EOS can still carry semantics of the entire prompt and influence its restoration. To resolve this, *STCFG* disables text conditioning entirely in ungrounded areas by using unconditional noise prediction in the reverse diffusion process, further enhancing the ungrounded region's restoration.

content where it does not belong. A similar issue is shown in Fig. 2 (third row), where the incorrect tag 'camouflage' is extracted by DAPE from the degraded LR image. As a result, the SeeSR baseline synthesizes camouflage-like patterns in regions (highlighted in red boxes) that should depict stones. Although the output appears visually plausible, it suffers from poor semantic fidelity.

**Incorrect prompts**. Extracting accurate text prompts from LR images remains challenging. Recent efforts, such as the degradation-aware prompt extractor (DAPE) [54], improve on the Recognize Anything Model (RAM) but still produce misguiding prompts under severe degradation. Injecting these incorrect signals can lead to even more erroneous SR results than using no text guidance at all.

**Incomplete prompts**. While DAPE demonstrates greater robustness to image degradations than many alternative text extractors, it does not guarantee comprehensive coverage of the entire image. It may miss salient objects and often fails to assign tags to non-object background regions due to its object-centric design. As a result, uncovered regions encounter two key issues: (1) they lack the semantic guidance that tagged regions benefit from, and (2) they remain susceptible to influence from unrelated text prompts, leading to hallucinations in areas that should remain unaffected.

In response, we propose a novel *Spatially Re-focused Super-Resolution (SRSR)* framework (Fig. 3) that comprises two key components. First, the *Spatially Re-focused Cross-Attention (SRCA)* module (Sec. 3.2) tackles the first two gaps by mitigating semantic ambiguity and reducing the influence of

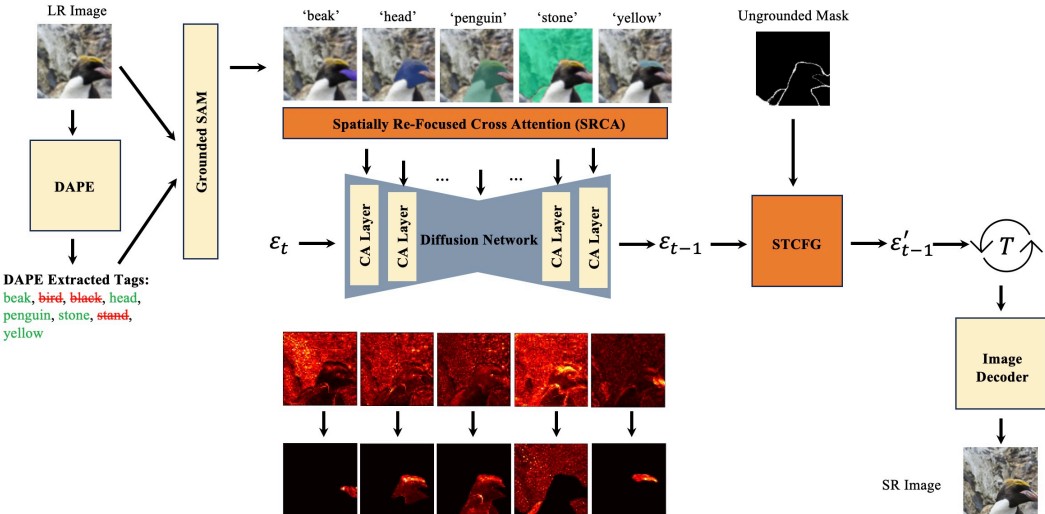

Figure 3: Overview of our proposed pipeline. First, the LR image is processed by a Degradation-Aware Prompt Extractor (DAPE) [54] to obtain text tags. Both the LR image and the extracted tags are then passed to Grounded SAM, which produces visually grounded tag–mask pairs. We also define an ungrounded mask as the complement of the union of all grounded masks. Next, each tag–mask pair is integrated into all 16 U-Net cross-attention layers, using the masks to constrain text conditioning precisely to relevant regions. Once noise prediction is complete, we selectively apply Classifier-Free Guidance (CFG) to grounded pixels while leaving ungrounded pixels under unconditional guidance, ensuring they remain unaffected by the text prompts. After $T$ denoising steps, a final decoder maps the latent representation back to pixel space, yielding the super-resolved (SR) image.

incorrect or loosely related tags through spatially constrained attention. Second, the *Spatially Targeted Classifier-Free Guidance (STCFG)* module (Sec. 3.3) targets the third limitation by improving restoration in regions lacking explicit grounding. Notably, SRSR is designed as an efficient, inference-only solution that requires no additional training or fine-tuning, making it compatible as a plug-and-play module with any cross-attention-based super-resolution method that utilizes text priors.

## 3.2 Spatially Re-focused Cross-Attention (SRCA)

First, we ground the DAPE-extracted tags onto the image using Grounded SAM [35], assigning each tag to a specific region. This step offers two key benefits: (1) tags that cannot be confidently grounded are likely irrelevant and thus removed, avoiding the injection of erroneous signals into the super-resolution process, addressing the aforementioned second limitation; (2) each grounded tag is now accompanied by a segmentation mask, enabling our proposed *Spatially Re-focused Cross-Attention (SRCA)* mechanism to mitigate semantic ambiguity and address the aforementioned first limitation.

With these text-mask pairs, SRCA refines the spatial precision of text conditioning, effectively reducing semantic ambiguity in the super-resolved outputs. Specifically, instead of only relying on the implicit cross-attention itself, we explicitly leverage each tag's corresponding segmentation mask to localize and re-focus the attention of text-conditioning, effectively suppressing attention outside the designated target regions. This ensures that each text token attends exclusively to its intended image regions, mitigating distractions from irrelevant areas. See Fig. 1 and 2 for visual analysis.

Formally, the cross-attention mechanism computes attention weights $\alpha_{ij}$ for pixel $i$ and token $j$ as:

$$\alpha_{ij} = \text{SoftMax}\left(\frac{Q_i \cdot K_j}{\sqrt{d}}\right), \tag{1}$$

where $Q$ and $K$ are query and key embeddings, respectively, and $d$ is the feature dimension. These weights form a weighted sum of the token values $V_j$ to obtain $O_i$, which is the output for pixel $i$:

$$O_i = \sum_j \alpha_{ij} V_j, \tag{2}$$

In SRCA, we first apply a binary mask $M_{ij}$ (1 for valid regions) to refine the attention weights $\alpha_{ij}$:

$$\alpha_{ij}^{\text{SRCA}} = M_{ij}\,\alpha_{ij}. \tag{3}$$

We then re-normalize these masked weights across both pixel and token dimensions:

$$\widehat{\alpha}_{ij}^{\text{SRCA}} = \frac{\alpha_{ij}^{\text{SRCA}}}{\sum_{i',j'} \alpha_{i'j'}^{\text{SRCA}}}. \tag{4}$$

This preserves proper attention distribution and ensures that zeroed-out entries do not distort the attention map while preserving the normalization of valid tokens, ensuring they continue to sum to one. As a result, this re-focusing operation suppresses contributions from irrelevant tokens, ensuring that each pixel $i$ attends only to the relevant tokens while maintaining consistent overall attention magnitudes. For example, for pixel $i$, a relevant token $V_m$ would correspond to $M_{im} = 1$, resulting in the attention weight unchanged $\alpha_{ij}^{\text{SRSR}} = \alpha_{ij}$. Conversely, for an irrelevant token $V_n$, the segmentation mask would have $M_{in} = 0$, resulting in no attention is placed on such a token. Importantly, as Eq. 2 computes weighted average of all tokens, eliminating such irrelevant token also helps increase the attentions to the relevant tokens. Consequently, SRCA produces more contextually aligned super-resolution results, with enhanced focus on the key regions of the image.

### 3.3 Spatially Targeted CFG application (STCFG)

Although our proposed *SRCA* in Sec. 3.2 addresses the first two limitations, the third limitation of insufficient tag coverage remains. Moreover, when our grounding mechanism is overly conservative and discards tags that are partially relevant, coverage can be reduced further. Resolving this issue is crucial for improving our overall performance. A recent work [40] proposed an interesting approach to address this challenge by employing Mask2Former [5], trained on ADE20K [70], to generate semantic segmentation labels for each image region, which are then used as text conditions. However, unlike DAPE, Mask2Former is not degradation-aware and often produces inaccurate labels when applied to degraded LR images, failing to resolve the second limitation and ultimately resulting in inferior performance. Furthermore, the model trained on ADE20K is limited to a fixed set of 150 categories, which restricts its expressiveness and precision in practice. Through additional ablation studies (Tab. 2), we observe that using these additional segmentation tags in the prompt leads to degraded performance. We also tested open-vocabulary segmentation methods, such as Prompt-Free Anything Detection and Segmentation in DINO-X [34], and observed that while they outperformed Mask2Former, their lack of degradation-awareness similarly led to inaccurate tags and inferior results compared to using DAPE alone. These indicate that inaccuracies are more harmful than incomplete coverage for text conditionings. More discussions are available in the ablation studies (Sec. 4.3).

Given the inherent difficulty and threshold sensitivity in balancing accuracy and coverage of tags extracted from degraded LR images, we note that the coverage limitation affects only the ungrounded regions. Instead of fine-tuning thresholds, can we explore an alternative that directly improves ungrounded-region quality by design without risking incorrect prompts? Drawing from our insight that erroneous tags are more detrimental than no tags, we argue that omitting guidance is preferable to providing incorrect conditioning. In standard CFG, a text-conditional output $\epsilon_\theta(x_t, y)$ is balanced against an unconditional output $\epsilon_\theta(x_t, \phi)$. The combined prediction $\hat{\epsilon}$ is typically computed as:

$$\hat{\epsilon} \leftarrow \epsilon_\theta(x_t, \phi) + s\big[\epsilon_\theta(x_t, y) - \epsilon_\theta(x_t, \phi)\big], \tag{5}$$

where $s \geq 1$ is the guidance scale, indicating a positive weight is applied to the text-conditional generation, while a non-positive weight is assigned to the unconditional generation. However, uniformly applying CFG can degrade ungrounded regions because they only receive prompts from semantic-free or summary global tokens (e.g., EOS, padding, punctuation), where summary tokens like EOS can still carry semantics of the entire prompt and influence its restoration. To address this, we construct an ungrounded mask $M$ for each image by taking the complement of the union of all grounded segmentation masks (i.e., $M = 1$ for ungrounded pixels and 0 for grounded pixels). We then *selectively* apply CFG only to grounded pixels, while leaving the ungrounded pixels unconditional:

$$\hat{\epsilon}_i \leftarrow (1 - M_i)\Big[\epsilon_\theta(x_t, \phi) + s\big(\epsilon_\theta(x_t, y) - \epsilon_\theta(x_t, \phi)\big)\Big] + M_i\,\epsilon_\theta(x_t, \phi), \tag{6}$$

where $M_i \in \{0, 1\}$ indicates whether pixel $i$ is ungrounded. This targeted approach ensures that uncertain or irrelevant tags do not influence the restoration of ungrounded regions, while grounded pixels still benefit from text conditioning, thereby preserving the visual fidelity.

Table 1: Quantitative comparison with state-of-the-art methods on both synthetic and real-world baselines. '$s$' denotes the number of diffusion reverse steps. Red = best, **Blue** = second-best.

| Dataset | Method | Metric | | | | | | | | |
|---|---|---|---|---|---|---|---|---|---|---|
| | | PSNR↑ | SSIM↑ | LPIPS↓ | DISTS↓ | FID↓ | NIQE↓ | MUSIQ↑ | MANIQA↑ | CLIPIQ↑ |
| RealSR | StableSR-s200 | 24.70 | 0.7085 | 0.3018 | 0.2288 | 128.51 | 5.9122 | 65.78 | 0.6221 | 0.6178 |
| | DiffBIR-s50 | 24.75 | 0.6567 | 0.3636 | 0.2312 | 128.99 | 5.5346 | 64.98 | 0.6246 | 0.6463 |
| | PASD-s20 | 25.21 | 0.6798 | 0.3380 | 0.2260 | 124.29 | 5.4137 | 68.75 | 0.6487 | 0.6620 |
| | ResShift-s15 | 26.31 | 0.7421 | 0.3460 | 0.2498 | 135.93 | 7.2635 | 58.43 | 0.5285 | 0.5444 |
| | SinSR-s1 | 26.28 | 0.7347 | 0.3188 | 0.2353 | 131.93 | 6.2872 | 60.80 | 0.5385 | 0.6122 |
| | OSEDiff-s1 | 24.43 | 0.7153 | 0.3173 | 0.2363 | 125.97 | 6.3897 | 67.52 | 0.6168 | 0.6742 |
| | SeeSR-s50 | 25.18 | 0.7216 | 0.3009 | 0.2223 | 125.55 | 5.4081 | 69.77 | 0.6442 | 0.6612 |
| | SRSR-OSEDiff-s1 | 24.53 | 0.7206 | 0.3166 | 0.2378 | 132.55 | 6.6106 | 67.24 | 0.6137 | 0.6710 |
| | SRSR-SeeSR-s50 | 26.40 | 0.7632 | 0.2718 | 0.2092 | 126.31 | 5.8627 | 62.88 | 0.5628 | 0.5409 |
| DIV2K-Val | StableSR-s200 | 23.26 | 0.5726 | 0.3113 | 0.2048 | 24.44 | 4.7581 | 65.92 | 0.6192 | 0.6771 |
| | DiffBIR-s50 | 23.64 | 0.5647 | 0.3524 | 0.2128 | 30.72 | 4.7042 | 65.81 | 0.6210 | 0.6704 |
| | PASD-s20 | 23.14 | 0.5505 | 0.3571 | 0.2207 | 29.20 | 4.3617 | 68.95 | 0.6483 | 0.6788 |
| | ResShift-s15 | 24.65 | 0.6181 | 0.3349 | 0.2213 | 36.11 | 6.8212 | 61.09 | 0.5454 | 0.6071 |
| | SinSR-s1 | 24.41 | 0.6018 | 0.3240 | 0.2066 | 35.57 | 6.0159 | 62.82 | 0.5386 | 0.6471 |
| | OSEDiff-s1 | 23.31 | 0.5970 | 0.3046 | 0.2129 | 26.80 | 5.4031 | 65.56 | 0.5857 | 0.6588 |
| | SeeSR-s50 | 23.68 | 0.6043 | 0.3194 | 0.1968 | 25.90 | 4.8102 | 68.67 | 0.6240 | 0.6936 |
| | SRSR-OSEDiff-s1 | 23.43 | 0.6023 | 0.3053 | 0.2142 | 27.65 | 5.5389 | 65.08 | 0.5797 | 0.6531 |
| | SRSR-SeeSR-s50 | 24.72 | 0.6416 | 0.3275 | 0.1991 | 25.31 | 5.4986 | 59.55 | 0.5402 | 0.5518 |
| DrealSR | StableSR-s200 | 28.03 | 0.7536 | 0.3284 | 0.2269 | 148.98 | 6.5239 | 58.51 | 0.5601 | 0.6356 |
| | DiffBIR-s50 | 26.71 | 0.6571 | 0.4557 | 0.2748 | 166.79 | 6.3124 | 61.07 | 0.5930 | 0.6395 |
| | PASD-s20 | 27.36 | 0.7073 | 0.3760 | 0.2531 | 156.13 | 5.5474 | 64.87 | 0.6169 | 0.6808 |
| | ResShift-s15 | 28.46 | 0.7673 | 0.4006 | 0.2656 | 172.26 | 8.1249 | 50.60 | 0.4586 | 0.5342 |
| | SinSR-s1 | 28.36 | 0.7515 | 0.3665 | 0.2685 | 170.57 | 6.9907 | 55.33 | 0.4884 | 0.6383 |
| | OSEDiff-s1 | 27.65 | 0.7743 | 0.3177 | 0.2366 | 141.96 | 7.3050 | 63.55 | 0.5758 | 0.7060 |
| | SeeSR-s50 | 28.17 | 0.7691 | 0.3189 | 0.2315 | 147.39 | 6.3967 | 64.93 | 0.6042 | 0.6804 |
| | SRSR-OSEDiff-s1 | 27.72 | 0.7781 | 0.3178 | 0.2379 | 139.66 | 7.5271 | 63.62 | 0.5736 | 0.7010 |
| | SRSR-SeeSR-s50 | 29.50 | 0.8128 | 0.2866 | 0.2176 | 146.98 | 7.1279 | 54.04 | 0.4962 | 0.5513 |

Visually, as shown in Fig. 2, SRCA and STCFG play complementary roles. SRCA improves the semantic fidelity of grounded objects by refining the cross-attention maps. In the top example, the SeeSR baseline mistakenly restores part of the building (highlighted in red) as water-like due to attention leakage from the 'water' token. SRCA re-focuses the attention of both 'building' and 'water' tokens to their correct regions, leading to the accurate restoration of the entire building. However, SRCA falls short in suppressing the influence of global tokens on ungrounded regions (as visualized by the ungrounded masks). To address this, STCFG applies unconditional guidance to those regions, effectively preventing hallucinations and yielding more faithful restorations, as qualitatively observed.

# 4 Experimental results

## 4.1 Setup

**Datasets.** Unlike prior approaches [44, 25, 58, 54, 53] that require training, our proposed SRSR operates purely at inference time and therefore only needs test datasets. Following the baselines, we adopt the standard test set from StableSR [44], which includes both synthetic and real-world data. The synthetic portion contains 3,000 images (512×512), whose ground truths (GT) are randomly cropped from DIV2K-Val [1] and degraded via the Real-ESRGAN pipeline [46]. The real-world portion comprises LQ-HQ pairs from RealSR [2] (128×128) and DRealSR [52] (512×512), respectively.

**Baseline methods.** SRSR is a plug-and-play module that integrates seamlessly into any cross-attention-based SR approach utilizing text priors, offering performance gains without additional training. To validate its effectiveness, we incorporate SRSR into two state-of-the-art methods: a 50-step method, SeeSR [54], and a single-step method, OSEDiff [53]. Users can select between these baselines based on their preferred trade-off between efficiency and quality, since multi-step approaches often yield better results at the expense of increased computation.

We further compare our SRSR-SeeSR and SRSR-OSEDiff variants against a range of strong baselines under their standard configurations, including StableSR [44], DiffBIR [25], PASD [58], ResShift [60], and SinSR [50]. Among these, ResShift trains a diffusion model from scratch in the pixel space, while SinSR is a one-step distilled model derived from ResShift. The remaining methods operate in latent space and build on top of the pre-trained SD model.

**Evaluation metrics.** In line with prior work, we evaluate using both full-reference and no-reference metrics. For full-reference evaluation, we use PSNR and SSIM [51] as fidelity measures, and LPIPS [66] and DISTS [7] to assess perceptual quality. FID [15] is also employed to gauge the distributional distance between super-resolved and ground-truth high-resolution (HR) images. For no-reference assessment, we adopt NIQE [64], MANIQA-pipal [57], MUSIQ [20], and CLIPIQA [43].

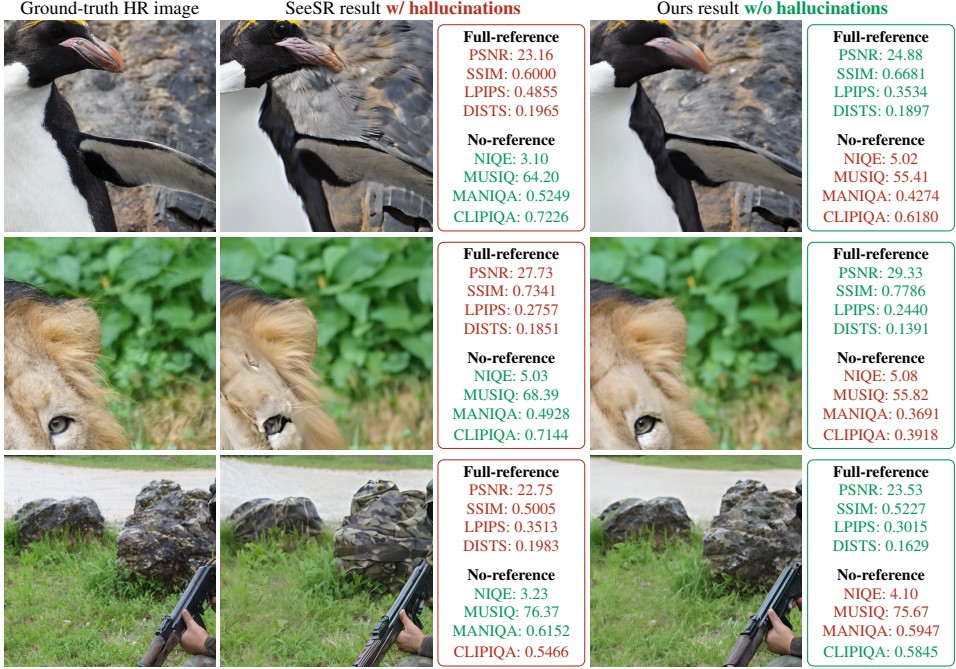

| Ground-truth HR image | SeeSR result **w/ hallucinations** | Ours result **w/o hallucinations** |

Figure 4: Qualitative results show that the baseline method exhibits clear semantic hallucinations. In contrast, plugging SRSR into the baseline leads to semantically faithful restorations. All full-reference metrics, including both fidelity (PSNR and SSIM) and perceptual quality (LPIPS and DISTS) measures, consistently validate the improvements brought by our method. However, it is worth noting that *the no-reference metrics tend to misjudge and heavily reward hallucinated results* due to their design, as indicated by the significant performance gap observed despite degraded semantic realism. This exposes their limitations in assessing semantic fidelity for super-resolution.

**Implementation details.** We adopt DAPE [54] as our prompt extractor, following its use in both SeeSR and OSEDiff. DAPE excels at handling degraded LR images while being more efficient than advanced multimodal vision-language models [26]. Although we experimented with various semantic segmentation models (e.g., Mask2Former [5] and Prompt-Free Anything Detection/Segmentation in DINO-X [34]), they did not make it into our final SRSR design. For grounding, we employ Grounded SAM 2, which is a combined framework built upon DINO-X and SAM 2.

**Complexity analysis.** Within our proposed SRSR framework, both SRCA and STCFG are applied only during inference and modify the cross-attention and CFG processes dynamically without introducing any new learnable parameters. Our approach uses the same pretrained diffusion model and UNet architecture as the baseline, so the parameter count remains the same. Therefore, the performance gains are achieved without increasing model complexity or adding new parameters.

**Efficiency analysis.** Our integration of Grounded SAM into the pipeline is highly efficient. *First*, it requires only a single inference per image before the SR process to pre-compute the masks, using the low-resolution (LR) input. The resulting masks are cached and reused during inference, avoiding repeated computation. *Second*, since only the LR image (e.g., 128×128) is fed into Grounded SAM, the processing time is minimal (just 0.12s per image) on V100, and even for 512×512 inputs, it remains low at 0.16s, making our approach practical and scalable for real-world applications. This pre-processing step also does not affect the parameter count or inference complexity.

## 4.2 Comparisons with state-of-the-art baselines

Tab. 1 compares our methods against several state-of-the-art baselines. To highlight SRSR's contribution, note how it improves a given baseline: for instance, applying SRSR to SeeSR considerably boosts full-reference fidelity measures PSNR and SSIM across all three datasets. It also enhances the full-reference perceptual quality metrics LPIPS and DISTS on two real-world sets (RealSR and DRealSR) while offering comparable performance on synthetic data (DIV2K-Val). No-reference metrics, however, often favor strong semantic cues, even if they are inaccurate, where it is frequently observed that hallucinated restorations have high performance in no-reference metrics, making them less reliable. Examples can be observed from Fig. 4. We also evaluate SRSR on OSEDiff,

which differs from SeeSR in two key aspects: it is a one-step inference method (confirming SRSR's flexibility regarding inference steps) and does not employ classifier-free guidance (CFG) (making STCFG inapplicable). In this setting, only the SRCA module is integrated. Despite this, we observe consistent improvements in fidelity metrics (PSNR and SSIM) across all datasets, with only marginal trade-offs in perceptual quality (LPIPS and DISTS), demonstrating SRCA's standalone effectiveness in enhancing semantic accuracy. This observation is further supported by the comparison between V2 and V3 in Tab. 2, which highlights the standalone effectiveness of SRCA without STCFG. The slight drop in perceptual quality arises from ungrounded regions still receiving text-conditioning from summary or meaningless tokens (e.g., EOS, punctuation), which STCFG is designed to address. This indicates that while SRSR is broadly compatible with diverse frameworks, its full potential is best realized when the underlying method supports CFG. Nonetheless, using SRCA alone is valuable in common scenarios where semantic fidelity is prioritized over background perceptual quality. These results highlight SRCA's independent benefits and STCFG's complementary role.

Beyond direct baselines, we have also compare with a range of strong competitors, where our methods generally achieve the best performance on full-reference fidelity and perceptual metrics, falling short of the top rank only in LPIPS and DISTS on DIV2K-Val, yet still securing second place in such cases.

## 4.3 Ablation studies

**Grounding.** As discussed in Sec. 3.2 and Sec. 3.3, grounding increases tag accuracy by discarding irrelevant text at the expense of reduced coverage when certain relevant tags are mistakenly removed. This reflects an accuracy-completeness trade-off in tag extraction. Experimentally, we find that accuracy typically outweighs completeness: providing fewer but more relevant tags produces better results than including extra but less reliable ones. For instance, comparing V1 and V2 in Tab. 2 shows that simply applying grounding (V2) to the SeeSR baseline yields small gains in three out of four metrics. Moreover, V5 differs from our final V4 only by including ungrounded tags, which lowers performance on all four metrics. Additionally, grounding's effectiveness depends on its confidence threshold, which balances strictness (accuracy) against coverage (completeness). We further explore this trade-off in Sec. 4.4. Finally, it is also worth emphasizing that grounding does more than just manage the accuracy-completeness trade-off: it also enables spatial re-focusing through SRCA, whereby each grounded tag is paired with a segmentation mask to alleviate semantic ambiguity.

**Spatially Re-focused Cross-Attention (SRCA).** SRCA mitigates semantic ambiguities and enhances output fidelity. When used alone without STCFG (V2 vs.V3), it improves fidelity at a slight cost to perceptual quality. However, when combined with STCFG (V4 vs.V6), SRCA yields improvements across all four metrics, underscoring its importance and complementary role within the full framework.

**Spatially Targeted CFG application (STCFG).** STCFG provides a direct mechanism for enhancing ungrounded regions and significantly improves all four metrics, whether used alone (V2 vs.V6), in combination with SRCA (V3 vs.V4), or in combination with both SRCA and different semantic segmentation methods (V9 vs.V7 and V10 vs.V8), demonstrating its effectiveness and versatility.

**DAPE vs. Semantic Segmentation.** As mentioned in Sec. 3.3, a contemporary work uses semantic segmentation (SS), specifically, Mask2Former [5] pre-trained on ADE20K, to extract semantic labels with complete image coverage. Motivated by their interesting work, we also experimented with

Table 2: Ablation study evaluating the contribution of each component within SRSR. SRCA denotes *spatially re-focused cross-attention*, and STCFG denotes *spatially targeted classifier-free guidance*. Mask2Former and DINO-X represent different semantic segmentation backbones. **Bold** indicates the best performance. Refer to Tab. S1 for the extended ablation results with no-reference metrics.

| Version | PSNR↑ | SSIM↑ | LPIPS↓ | DISTS↓ |
|---|---|---|---|---|
| V1: SeeSR | 25.1717 | 0.7219 | 0.3008 | 0.2223 |
| V2: SeeSR + Grounding | 25.1751 | 0.7234 | 0.3001 | 0.2229 |
| V3: SeeSR + Grounding + SRCA | 25.2688 | 0.7280 | 0.3013 | 0.2254 |
| V4: SeeSR + Grounding + SRCA + STCFG (Ours) | **26.3996** | **0.7632** | **0.2718** | 0.2092 |
| V5: SeeSR + Grounding + SRCA + STCFG + Ungrounded Tags | 26.3871 | 0.7625 | 0.2729 | 0.2095 |
| V6: SeeSR + Grounding + STCFG | 26.3986 | 0.7627 | 0.2735 | 0.2112 |
| V7: SeeSR + Grounding + SRCA + STCFG + Mask2Former | 26.3128 | 0.7620 | 0.2725 | 0.2093 |
| V8: SeeSR + Grounding + SRCA + STCFG + DINO-X | 26.3449 | 0.7627 | 0.2722 | **0.2089** |
| V9: SeeSR + Grounding + SRCA + Mask2Former | 26.2885 | 0.7609 | 0.2734 | 0.2098 |
| V10: SeeSR + Grounding + SRCA + DINO-X | 26.3221 | 0.7621 | 0.2729 | 0.2090 |

augmenting the well-performing DAPE tags using SS. However, as shown by comparing V4 and V7 in Tab. 2, performance declined, primarily because SS models, not trained with degradation in design, often produce inaccurate labels. Moreover, Mask2Former recognizes only 150 categories, limiting coverage for real-world SR tasks. This further reduce the semantic accuracy of the extracted texts.

To mitigate these constraints, we tried Prompt-Free Anything Detection and Segmentation in DINO-X [34] (V8 in Tab. 2), which supports open-vocabulary SS. Although it outperforms Mask2Former (V7), it still trails behind using only DAPE (V4). These results reaffirm that inaccurate tags prove more detrimental than incomplete ones - an insight that also reinforces our grounding design.

### 4.4 Hyper-parameter analysis

The confidence threshold for grounding affects the balance between tag accuracy and coverage. A higher threshold discards less certain tags to boost accuracy, but also reduces coverage. As shown in Table 3, larger thresholds generally yield higher PSNR and SSIM, whereas thresholds of 0.25-0.35 achieve more favorable LPIPS and DISTS. Overall, SRSR remains robust against varying thresholds: in all cases, it significantly outperforms the baseline in all four metrics. We adopt a confidence threshold of 0.25 for all reported comparisons.

Table 3: Hyper-parameter analysis: different grounding thresholds produce consistent results that outperform the SeeSR baseline on RealSR dataset. **Red** = best, **Blue** = second-best.

| Threshold | PSNR↑ | SSIM↑ | LPIPS↓ | DISTS↓ |
|---|---|---|---|---|
| Baseline | 25.1717 | 0.7219 | 0.3008 | 0.2223 |
| 0.15 | 26.3632 | 0.7625 | 0.2727 | **0.2093** |
| 0.25 | 26.3996 | 0.7632 | **0.2718** | **0.2092** |
| 0.35 | 26.4550 | 0.7644 | **0.2715** | 0.2099 |
| 0.45 | **26.5378** | **0.7657** | 0.2731 | 0.2112 |
| 0.55 | **26.6008** | **0.7665** | 0.2735 | 0.2124 |

## 5 Conclusion

We introduced a *spatially re-focused super-resolution (SRSR)* framework to improve the semantic accuracy of text-conditioned, diffusion-based SR. By grounding and filtering extracted tags, we remove irrelevant text prompts and pair each valid tag with a corresponding segmentation mask. This enables *spatially re-focused cross-attention* (SRCA), which directs guidance to target regions, and *spatially targeted classifier-free guidance* (STCFG), which preserves visual fidelity in ungrounded areas. Our approach operates entirely at inference time and can be applied to any cross-attention-based SR model that leverages text priors. Experiments on both synthetic and real-world datasets show that SRSR outperforms seven state-of-the-art methods in fidelity and perceptual quality metrics.

## 6 Limitations and future works

While SRSR achieves strong gains in both fidelity and semantic accuracy, several limitations and future directions remain. *First*, further exploration of robust, degradation-aware segmentation models could reduce incomplete tag coverage and enhance tagging accuracy. Indeed, our ablation studies that compare Mask2Former with DINO-X suggest that stronger semantic segmentation (SS) backbones improve SRSR's performance, indicating that future innovations in SS can further enhance our method. *Second*, SRSR is currently designed as an inference-time framework to ensure plug-and-play compatibility with various pre-trained super-resolution models. Although this design enables broad applicability without retraining, incorporating SRSR into the training or fine-tuning process represents an exciting future direction. Doing so would allow the model to internalize the spatial and semantic constraints imposed by SRSR, potentially improving region-specific supervision and semantic localization. However, this integration would come at the cost of additional training complexity and resource demands. *Third*, SRSR's modular nature allows flexible integration with other Real-ISR frameworks. In this work, we demonstrated its generality through consistent improvements when applied to SeeSR [54] and OSEDiff [53]. Future research could further validate this generality by extending SRSR to broader architectures such as AddSR [56] and TSD-SR [10]. *Finally*, existing no-reference quality metrics remain unreliable, as they often over-reward artifact-heavy or hallucinated outputs (see Fig. 4 and Figures S1–S11). Hence, we emphasize full-reference fidelity and perceptual quality metrics as more trustworthy indicators of restoration faithfulness, which is critical in practical scenarios (e.g., e-commerce product imaging) where visual accuracy directly impacts trust and usability. Nevertheless, when high-quality references are unavailable, the dependence on current no-reference metrics becomes problematic. Developing hallucination-aware no-reference metrics that better align with full-reference measures and human perception is an urgent research need. Promising directions include leveraging vision-language models for perceptual assessment and incorporating penalties for semantic hallucinations.

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

## Supplementary Material

**Outline of supplementary material.** This supplementary material provides additional results, analyses, and discussions to support the main paper. It is structured as follows:

- In Section A, we present extensive qualitative comparisons and visual analyses by comparing the baseline results and cross-attention maps before and after integrating our proposed SRSR framework. These examples demonstrate the effectiveness and internal mechanisms of the SRCA and STCFG components within the SRSR framework (Figures S1–S11).

- Section B extends this analysis to an ablation variant that uses semantic segmentation masks for prompt extraction. This supplements the quantitative ablation study results in Section 4.3 of the main paper and supports the claim in Section 3.3 that semantic segmentation has two key limitations (lack of degradation awareness and constrained vocabulary size), which lead to inferior results compared to our approach based on DAPE and Grounded SAM (Figures S12–S15).

- In Section C, we provide additional qualitative comparisons paired with quantitative metrics to highlight a key observation: no-reference metrics often misjudge and favor hallucinated outputs (Figures S16–S17).

- Section D presents an additional user study evaluating human perceptual preferences between the baseline results and those produced after integrating SRSR (Figure S18). The study shows that human annotators consistently prefer our SRSR-enhanced outputs.

- Section E supplements the ablation results in Tab.2 with no-reference metrics (Tab. S1), further showing that while full-reference fidelity and perceptual metrics align with visual quality improvements, existing no-reference metrics often produce misleading trends and over-reward artifact-heavy outputs.

- Finally, Section F provides a clarification on potential societal impacts.

All figure references include detailed captions to support discussion and analysis of the corresponding findings.

## A    Additional qualitative comparisons and visual analysis

As a plug-and-play module, the effectiveness of SRSR is most clearly demonstrated by comparing the performance of the same baseline before and after integration. In this section, we present additional qualitative comparisons and visual analyses (Figures S1–S11) to support the following key takeaways:

1. **Enhanced semantic fidelity.** We provide qualitative comparisons that highlight the ability of SRSR to improve semantic fidelity by removing hallucinations. *While baseline results may appear visually appealing, they often contain hallucinated objects or textures that are semantically inconsistent with the ground-truth high-resolution image*. In contrast, our method produces restorations that better align with the true image content.

2. **Limitations of no-reference metrics.** Alongside visual results, we include corresponding quantitative comparisons. We emphasize that *no-reference metrics, which lack access to the ground-truth high-resolution image, tend to misjudge and often reward hallucinated outputs*. This is evident in the performance drop of these metrics when hallucinations are removed. Conversely, full-reference metrics (PSNR and SSIM for fidelity, and LPIPS and DISTS for perceptual quality) provide a more reliable assessment of semantic and visual accuracy. SRSR demonstrates superior performance in these full-reference evaluations.

3. **Visual analysis of SRCA and STCFG effects.** To further illustrate the mechanisms of our proposed components, we include auxiliary visualizations such as the re-focused versus original cross-attention maps or the ungrounded region masks. These demonstrate how *SRCA enhances text-token alignment with relevant regions* and how *STCFG effectively suppresses text influence in ungrounded areas to prevent hallucinations*.

All figures presented in this section are accompanied by detailed captions to support discussion and analysis of the corresponding findings.

## B    Comparing prompt extraction methods: DAPE vs. Semantic Segmentation

As discussed in Section 3.3 of the main paper, we also experimented with using semantic segmentation to extract grounded tag–region pairs, and compared its performance with our final method that employs DAPE and Grounded SAM for the same goal. We observed two key limitations when using semantic segmentation compared to DAPE: *First*, unlike DAPE, which is trained to be degradation-aware when extracting tags from low-quality LR images, standard semantic segmentation models lack this capability and tend to extract incorrect tags more frequently, leading to restoring hallucinated contents that are not faithful to the ground-truth HR image. *Second*, Semantic segmentation models often have a limited vocabulary size, which restricts their expressiveness and precision in practice. This leads to the extraction of coarse prompts that result in imprecise or incorrect restorations. These issues result in inferior performance compared to our final approach, which uses DAPE for prompt extraction followed by Grounded SAM for visual grounding.

As reported in Section 4.3 of the main paper, we already presented ablation studies with quantitative comparisons. In this section, we supplement those results with additional qualitative comparisons and analysis. Specifically, the figures correspond to supporting the following key takeaways:

1. **Semantic segmentation models lack degradation awareness.** Figures S12–S13 highlight that this limitation *causes incorrect prompts to be confidently extracted*, which then misguide the restoration process and lead to semantically incorrect outputs.

2. **Semantic segmentation models have constrained vocabulary size.** Figures S14–S15 illustrate that this limitation *reduces the expressiveness and precision of prompt extraction*. As a result, using only coarsely related prompts to guide the restoration process leads to suboptimal outcomes.

3. **Limitations of no-reference metrics.** Same as in Section A, all figures in this section (Figures S12–S15) further demonstrate the drawbacks of no-reference metrics: *they frequently misjudge and favor outputs that contain hallucinations resulting from incorrect text conditioning*, demonstrating their limitations in assessing semantic fidelity for super-resolution.

All figures presented in this section are also accompanied by detailed captions to support discussion and analysis of the corresponding findings.

## C    More qualitative comparisons illustrating no-reference metrics' limitations

The previous two sections have already highlighted the limitations of no-reference metrics by comparing our method against both the baseline and an ablation variant that uses semantic segmentation masks for prompt extraction. In this section (Figures S16–S17), we provide additional visual comparisons (paired with corresponding quantitative metrics) between our method and other baseline approaches to further underscore the shortcomings of no-reference metrics.

## D    Additional user study results

We have also conducted a user study to evaluate human perceptual preferences between the baseline results before and after integrating our SRSR framework. Given the size of the synthetic DIV2K dataset, we focus the study on two real-world datasets: RealSR and DRealSR, using their entire datasets. For each image, we presented the low-resolution input along with two anonymized high-resolution outputs (ours: SRSR-SeeSR and the baseline: SeeSR) labeled 'Method A' and 'Method B'. Two annotators were asked to select the preferred image or indicate if both appeared visually similar, with evaluations based on sharpness, visual realism, and detail preservation. The order of presentation was randomized, and full images were shown (not crops) to avoid bias. Results are aggregated and visualized in bar charts (Figure S18), showing that SRSR-SeeSR is consistently preferred over the baseline SeeSR across both datasets.

Table S1: Ablation study evaluating the contribution of each component within SRSR. SRCA denotes *spatially re-focused cross-attention*, STCFG denotes *spatially targeted classifier-free guidance*, and G refers to grounding. Mask2Former and DINO-X represent different semantic segmentation backbones. **Bold** indicates the best performance.

| Version | PSNR↑ | SSIM↑ | LPIPS↓ | DISTS↓ | FID↓ | NIQE↓ | MUSIQ↑ | MANIQA↑ | CLIPIQA↑ |
|---|---|---|---|---|---|---|---|---|---|
| V1: SeeSR | 25.1717 | 0.7219 | 0.3008 | 0.2223 | **125.55** | **5.4081** | 69.77 | **0.6442** | 0.6612 |
| V2: SeeSR + G | 25.1751 | 0.7234 | 0.3001 | 0.2229 | 128.41 | 5.4852 | 69.84 | 0.6436 | **0.6701** |
| V3: SeeSR + G + SRCA | 25.2688 | 0.7280 | 0.3013 | 0.2254 | 132.55 | 5.5029 | **69.91** | 0.6422 | 0.6683 |
| V4: SeeSR + G + SRCA + STCFG (Ours) | **26.3996** | **0.7632** | **0.2718** | 0.2092 | 126.31 | 5.8627 | 62.88 | 0.5628 | 0.5409 |
| V5: V4 + Ungrounded Tags | 26.3871 | 0.7625 | 0.2729 | 0.2095 | 126.28 | 5.8657 | 63.15 | 0.5665 | 0.5460 |
| V6: SeeSR + G + STCFG | 26.3986 | 0.7627 | 0.2735 | 0.2112 | 127.24 | 5.8947 | 62.99 | 0.5648 | 0.5456 |
| V7: SeeSR + G + SRCA + STCFG + Mask2Former | 26.3128 | 0.7620 | 0.2725 | 0.2093 | 127.51 | 5.8026 | 63.31 | 0.5689 | 0.5477 |
| V8: SeeSR + G + SRCA + STCFG + DINO-X | 26.3449 | 0.7627 | 0.2722 | **0.2089** | 127.77 | 5.8196 | 63.17 | 0.5670 | 0.5450 |
| V9: SeeSR + G + SRCA + Mask2Former | 26.2885 | 0.7609 | 0.2734 | 0.2098 | 127.19 | 5.7801 | 63.73 | 0.5728 | 0.5543 |
| V10: SeeSR + G + SRCA + DINO-X | 26.3221 | 0.7621 | 0.2729 | 0.2090 | 127.33 | 5.7927 | 63.58 | 0.5713 | 0.5520 |

# E  Extended ablation results

To further strengthen our point that the existing no-reference metrics are flawed that they tend to over-reward artifact-heavy or hallucinated outputs, in Tab. S1, we supplement the no-reference metrics to the ablation results in Tab. 2.

Tab. S1 highlights a consistent trend: improvements in full-reference fidelity metrics (PSNR, SSIM) and perceptual quality metrics (LPIPS, DISTS) align with observed improvements in visual and semantic fidelity, as supported by our qualitative analysis (e.g., Fig. 4 and Figures S1–S11). However, the same changes often lead to worse results in all no-reference metrics (NIQE, MUSIQ, MANIQA, CLIPIQA), which can misleadingly favor outputs with hallucinated or artifact-heavy details. For example, removing STCFG (V3) from our full version (V4) improves all no-reference metrics, even though fidelity, full-reference perceptual quality, and visual quality drop. Similarly, removing SRCA (V6) worsens both fidelity and full-reference perceptual quality, yet again, no-reference metrics do not consistently reflect this decline. This pattern also holds when comparing DINO-X to Mask2Former (V8/V7, V10/V9): fidelity, full-reference perceptual quality metrics, and visual quality are better with DINO-X, which aligns with the inherent advantages of DINO-X over Mask2Former, as we elaborated in Sec. 4.3. However, no-reference metrics show the reverse trend. These findings reinforce our point that current no-reference metrics often fail to penalize hallucinations and can mislead practical evaluation, especially in Real-ISR.

# F  Clarification on potential negative societal impact

Our work focuses on enhancing the visual quality of low-resolution images through super-resolution techniques with improved semantic fidelity. It is primarily intended for applications in image restoration, photography, and scientific imaging, where enhancing degraded real-world content is valuable. Our method does not involve any user or personally identifiable information, and our evaluation is conducted on public, non-sensitive datasets. Therefore, we believe our work does not pose significant societal risks. On the contrary, it may benefit the aforementioned intended applications where detail-preserving enhancement is crucial.

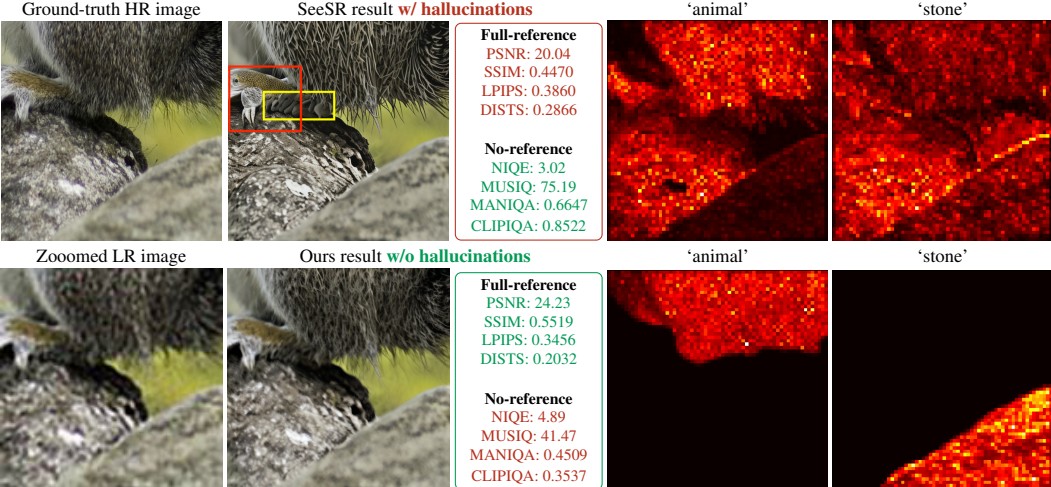

Figure S1: Column 1: Paired LR-HR images. Column 2: Qualitative comparisons of the baseline with and without our proposed SRSR. Column 3: Corresponding quantitative metrics. Columns 4–5: Attention visualizations for selected tokens to illustrate the effect of SRSR. In the region highlighted by the red bounding box, the object is actually an animal's claw, but it is difficult to recognize in the degraded LR image. Consequently, the prompt extractor (DAPE) fails to extract relevant tags such as 'claw'. As a result, the baseline model attributes this region to the token 'animal' according to the cross-attention map, and hallucinates it as a vivid fish. Similarly, in the yellow bounding box, the region corresponds to the animal's fur, but is misinterpreted due to degradation. The baseline instead applies influence from the unrelated tag 'stone', resulting in a texture resembling small pebbles. Beyond these highlighted objects, the baseline also introduces other over-synthesized textures in the 'animal' and 'stone' regions that deviate from the ground-truth HR image, despite being visually plausible. In contrast, our SRSR framework assigns tags only to regions where grounding confidence is high (e.g., 'animal', 'stone'), leaving uncertain regions, such as the red region, ungrounded. Within the SRSR framework, SRCA ensures that grounded regions are not influenced by irrelevant tokens, thereby removing hallucinations from the 'animal' and 'stone' areas. Additionally, STCFG applies unconditional predictions to ungrounded regions like the animal's claw, suppressing inappropriate text influence while preserving perceptual quality.

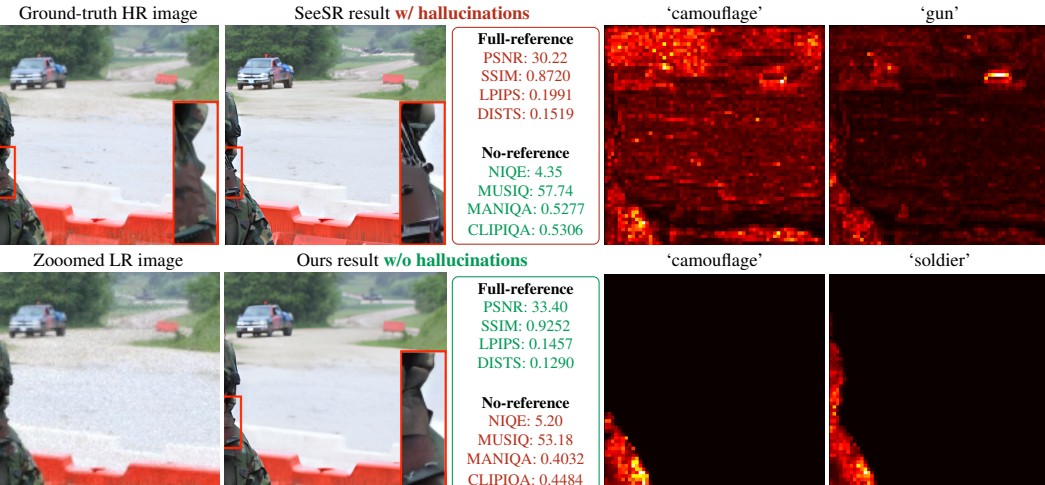

Figure S2: Column 1: Paired LR-HR images. Column 2: Qualitative comparisons of the baseline with and without our proposed SRSR. Column 3: Corresponding quantitative metrics. Columns 4–5: Attention visualizations for selected tokens to illustrate the effect of SRSR. In the highlighted region, the baseline hallucinates gun-like objects due to incorrect tags like 'gun' and 'rifle' extracted by DAPE, which misattribute attention to parts of the 'camouflage' object. In contrast, our SRSR framework removes such irrelevant tags via grounding and re-focuses attention on salient concepts like 'camouflage' and 'soldier', resulting in semantically faithful restorations.

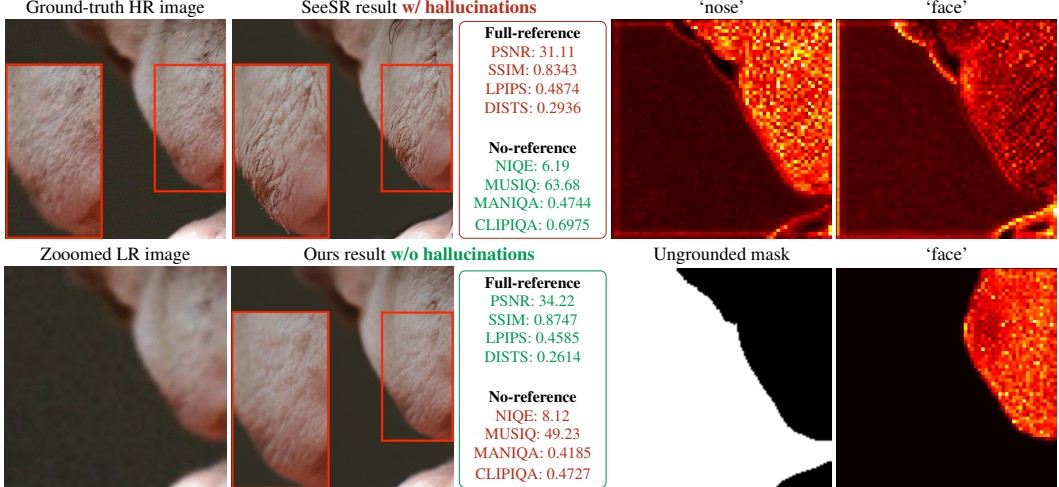

Figure S3: Column 1: Paired LR-HR images. Column 2: Qualitative comparisons of the baseline with and without our proposed SRSR. Column 3: Corresponding quantitative metrics. Columns 4-5: Ungrounded region mask and attention visualizations for selected tokens to illustrate the effect of SRSR. In the highlighted region, the baseline hallucinates nose hairs on the face due to incorrect tags like 'nose' and 'mouth' extracted by DAPE, which misattribute attention to parts of the 'face' object. In contrast, our SRSR framework removes such irrelevant tags via grounding and re-focuses attention on salient concepts like 'face', resulting in semantically faithful restorations.

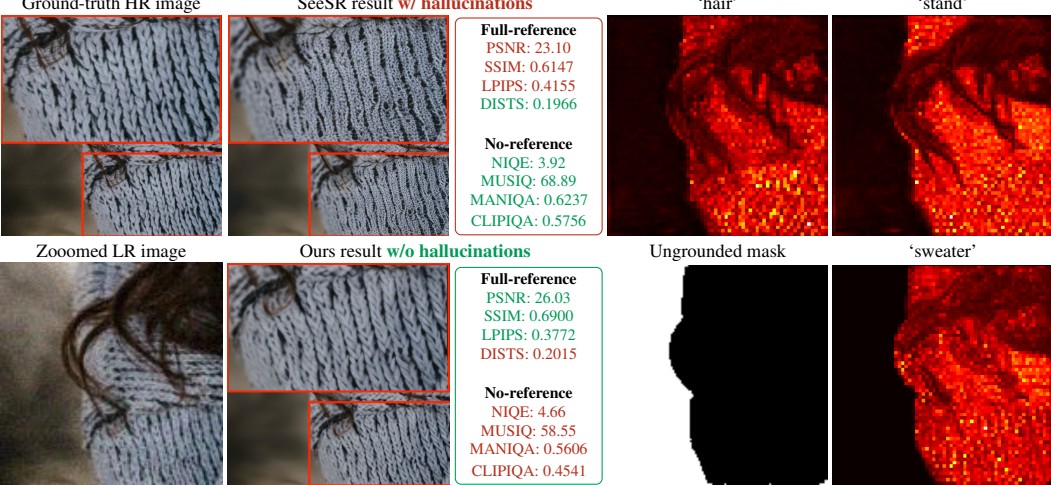

Figure S4: Column 1: Paired LR-HR images. Column 2: Qualitative comparisons of the baseline with and without our proposed SRSR. Column 3: Corresponding quantitative metrics. Columns 4-5: Ungrounded region mask and attention visualizations for selected tokens to illustrate the effect of SRSR. In the highlighted region, the baseline produces sweater textures that deviate from the ground-truth HR image, influenced by unrelated tags such as 'hair' and incorrect tags like 'stand', as indicated by the cross-attention maps. In contrast, our SRSR framework employs SRCA to eliminate irrelevant tag influence and localize text conditioning strictly to grounded regions, associating the sweater region only with the tag 'sweater', thereby ensuring more faithful and semantically accurate restorations.

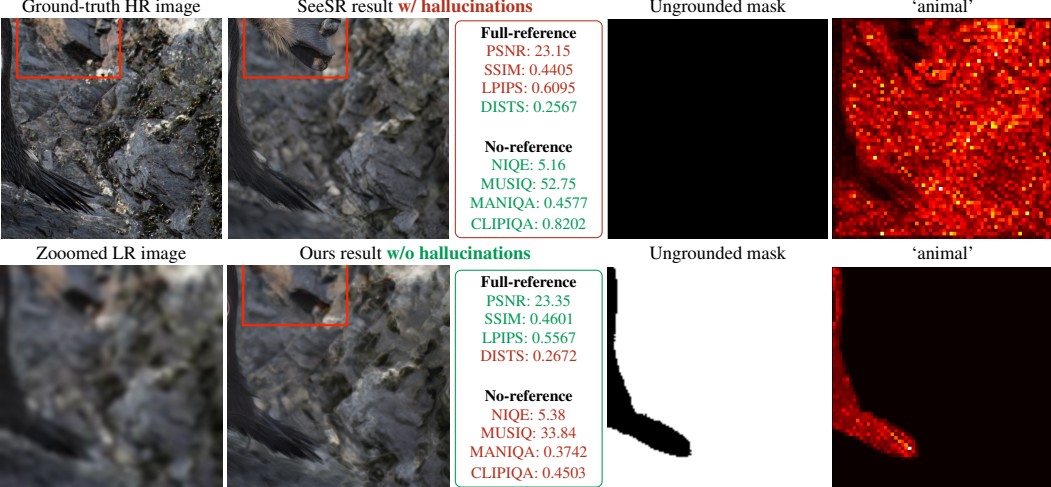

Figure S5: Column 1: Paired LR-HR images. Column 2: Qualitative comparisons of the baseline with and without our proposed SRSR. Column 3: Corresponding quantitative metrics. Columns 4-5: Ungrounded region mask and attention visualizations for selected tokens to illustrate the effect of SRSR. In the baseline, the inherent cross-attention for the token 'animal' is broadly dispersed across the image, including the highlighted region corresponding to the stone object, leading to hallucinated animal-like features in that area (note the stone is mistakenly restored with fur-like textures). Our SRSR framework addresses this by first applying SRCA to constrain the token 'animal' to its grounded region, correcting misaligned attention. However, ungrounded regions remain susceptible to influence from global tokens such as EOS, padding, and punctuation, which can introduce noise and semantic summary of the entire prompt. To address this, we introduce STCFG to explicitly avoid applying text conditioning to ungrounded regions, resulting in more semantically faithful restorations.

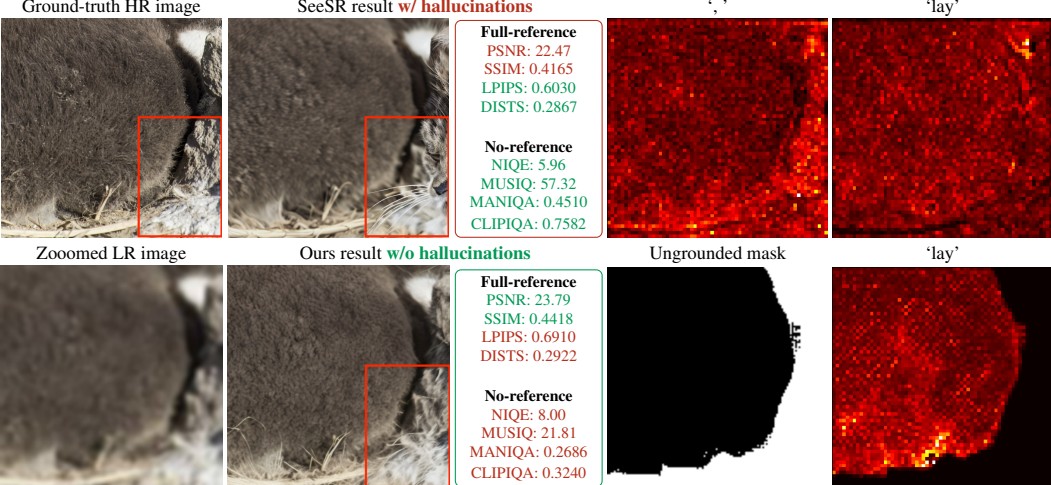

Figure S6: Column 1: Paired LR-HR images. Column 2: Qualitative comparisons of the baseline with and without our proposed SRSR. Column 3: Corresponding quantitative metrics. Columns 4-5: Ungrounded region mask and attention visualizations for selected tokens to illustrate the effect of SRSR. The highlighted region is heavily degraded and should depict stone and weed in the ground-truth HR image. However, the baseline incorrectly restores it as a cat's face with whiskers, which is visually plausible but semantically inaccurate. The baseline's cross-attention maps reveal that the punctuation token ', ' and the irrelevant token 'lay' attend to this region. These lack meaningful semantics or relevance to the scene, resulting in hallucinated content during restoration. By contrast, our SRSR framework applies SRCA to constrain the influence of tokens like 'lay' to their grounded regions and uses STCFG to apply unconditional generation in ungrounded areas (as identified by the ungrounded mask), effectively suppressing hallucinations and improving semantic fidelity.

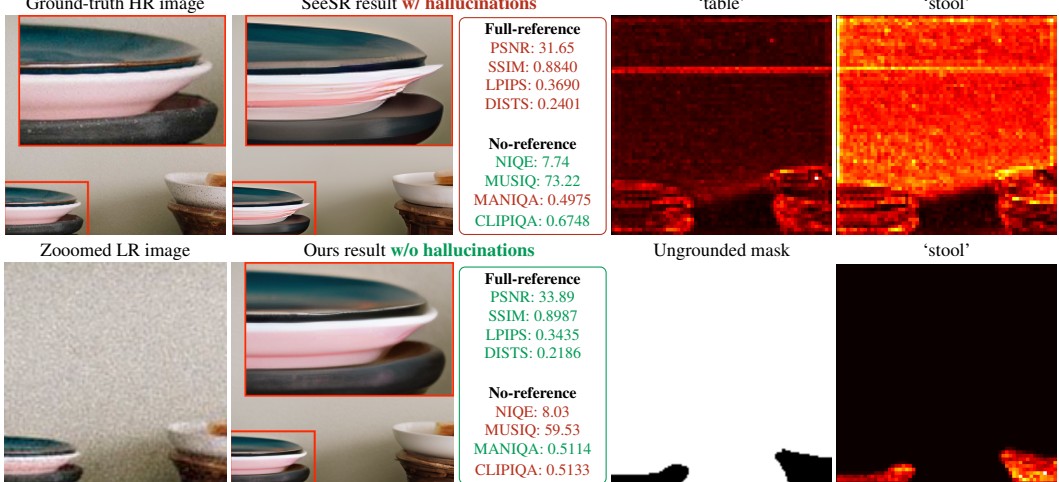

Figure S7: Column 1: Paired LR-HR images. Column 2: Qualitative comparisons of the baseline with and without our proposed SRSR. Column 3: Corresponding quantitative metrics. Columns 4-5: Ungrounded region mask and attention visualizations for selected tokens to illustrate the effect of SRSR. The key objects in this example are two stacks of plates, each placed on a supporting base. While DAPE successfully extracts the semantically related tag 'stool' for the base structures, it fails to recognize the plates due to severe degradation in the LR image. In the baseline, the cross-attention for the tag 'stool' is diffusely distributed across irrelevant regions, and additional tags such as 'table' and 'stool' incorrectly attend to the plate areas. This misattribution results in visible artifacts in the restored plates and suboptimal reconstruction of the base. In contrast, our SRSR framework leverages SRCA to re-focus attention on correctly grounded regions (e.g., the base) and uses STCFG to suppress text-based influence in ungrounded regions (e.g., the plates), yielding more accurate and faithful restorations.

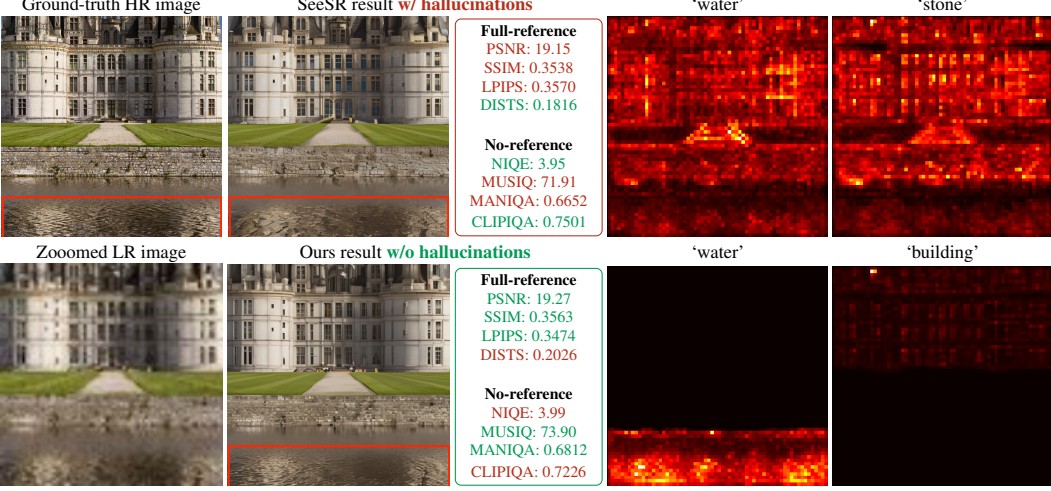

Figure S8: Column 1: Paired LR-HR images. Column 2: Qualitative comparisons of the baseline with and without our proposed SRSR. Column 3: Corresponding quantitative metrics. Columns 4–5: Attention visualizations for selected tokens to illustrate the effect of SRSR. In the highlighted water regions, the baseline introduces unnatural patterns, influenced by the irrelevant token 'stone' and the misdirected attention of the relevant token 'water' to incorrect spatial areas. By contrast, our SRSR framework leverages SRCA to re-focus each token's influence within its grounded spatial region, producing semantically more faithful and visually accurate restorations.

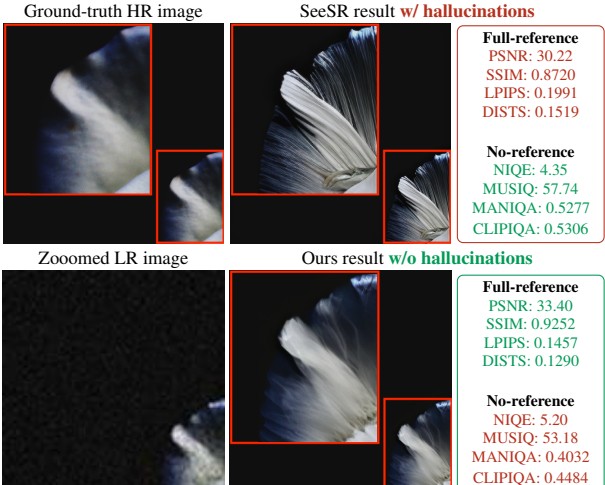

Figure S9: Column 1: Paired LR-HR images. Column 2: Qualitative comparisons of the baseline with and without our proposed SRSR. Column 3: Corresponding quantitative metrics. In this example, DAPE fails to extract any meaningful tags, leaving only global tokens (e.g., SOS, EOS, punctuation) to drive text conditioning. The baseline forces the application of text guidance even in such cases, causing the model to hallucinate textures despite the prompt carrying no semantic value. In contrast, our SRSR framework uses STCFG to disable text conditioning in ungrounded regions (here, the entire image), applying only unconditional prediction during denoising. This suppresses hallucinated details and improves alignment with the ground-truth HR image.

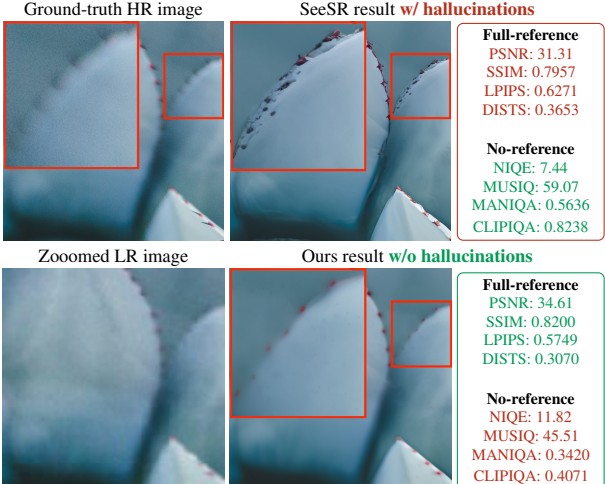

Figure S10: Column 1: Paired LR-HR images. Column 2: Qualitative comparisons of the baseline with and without our proposed SRSR. Column 3: Corresponding quantitative metrics. In this example, DAPE fails to extract any meaningful tags, leaving only global tokens (e.g., SOS, EOS, punctuation) to drive text conditioning. The baseline forces the application of text guidance even in such cases, causing the model to hallucinate textures despite the prompt carrying no semantic value. In contrast, our SRSR framework uses STCFG to disable text conditioning in ungrounded regions (here, the entire image), applying only unconditional prediction during denoising. This suppresses hallucinated details and improves alignment with the ground-truth HR image.

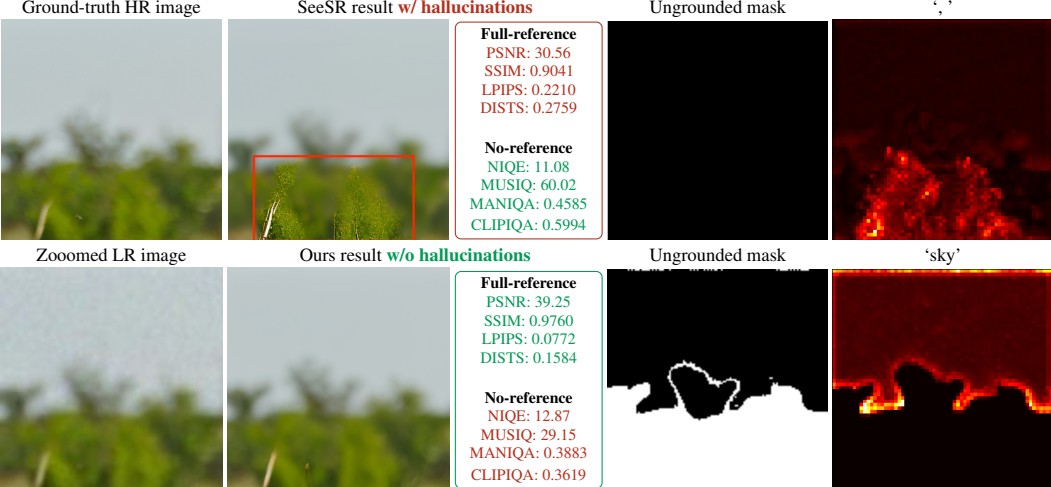

Figure S11: Column 1: Paired LR-HR images. Column 2: Qualitative comparisons of the baseline with and without our proposed SRSR. Column 3: Corresponding quantitative metrics. Columns 4-5: Ungrounded region mask and attention visualizations for selected tokens to illustrate the effect of SRSR. In the baseline, the inherent cross-attention maps show that only the punctuation token ', ' attends to the highlighted region. However, this token carries no meaningful semantics, and its influence during the generation process introduces noise and hallucinated details that deviate from the ground-truth HR image. In contrast, our SRSR framework leverages STCFG to apply unconditional generation specifically to regions that cannot be confidently grounded (i.e., those indicated by the ungrounded mask), effectively removing such hallucinations and improving semantic fidelity.

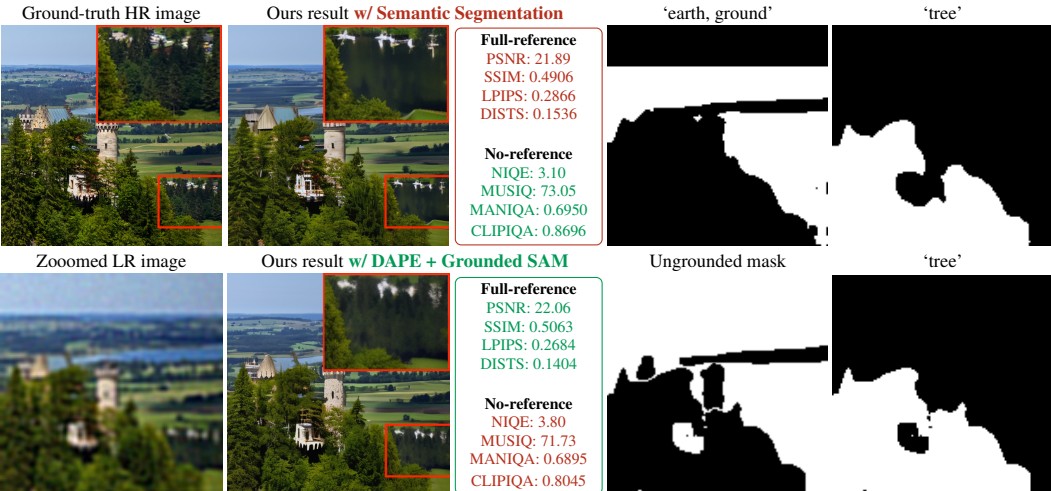

Figure S12: Column 1: Paired LR-HR images. Column 2: Qualitative comparisons of our results using semantic segmentation versus DAPE + Grounded SAM for prompt extraction and grounding. Column 3: Corresponding quantitative metrics. Columns 4–5: Grounded masks for selected tokens, where white indicates grounded regions. Unlike DAPE, which is trained to be degradation-aware when extracting tags from low-quality LR images, standard semantic segmentation models lack this capability and tend to extract incorrect tags more frequently. In the highlighted region, the semantic segmentation model confidently grounds the tag 'earth, ground', leading to a ground-like restoration that is inconsistent with the true content (trees) seen in the HR image. In contrast, DAPE is more degradation-aware, thus fails to confidently ground any tag for this region due to the severe degradation, triggering our STCFG mechanism to apply unconditional generation. This results in more faithful tree-like restoration and effectively prevents the introduction of semantically incorrect content.

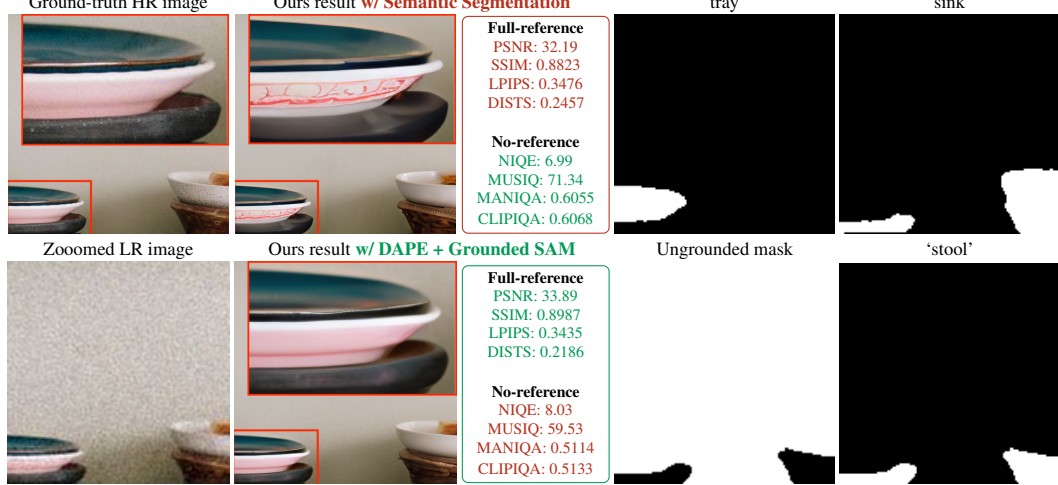

Figure S13: Column 1: Paired LR-HR images. Column 2: Qualitative comparisons of our results using semantic segmentation versus DAPE + Grounded SAM for prompt extraction and grounding. Column 3: Corresponding quantitative metrics. Columns 4–5: Grounded masks for selected tokens, where white indicates grounded regions. Unlike DAPE, which is trained to be degradation-aware when extracting tags from low-quality LR images, standard semantic segmentation models lack this capability and tend to extract incorrect tags more frequently. In the highlighted region, the semantic segmentation model grounds tags such as 'tray' and 'sink', which leads to hallucinated content — misrepresenting both the plates and the base beneath them. In contrast, DAPE extracts the tag 'stool', which more appropriately describes the supporting base. For the severely degraded plate regions, DAPE does not assign any confident tag, triggering our STCFG mechanism to apply unconditional generation. This helps produce more semantically accurate restorations and avoids introducing misleading visual artifacts.

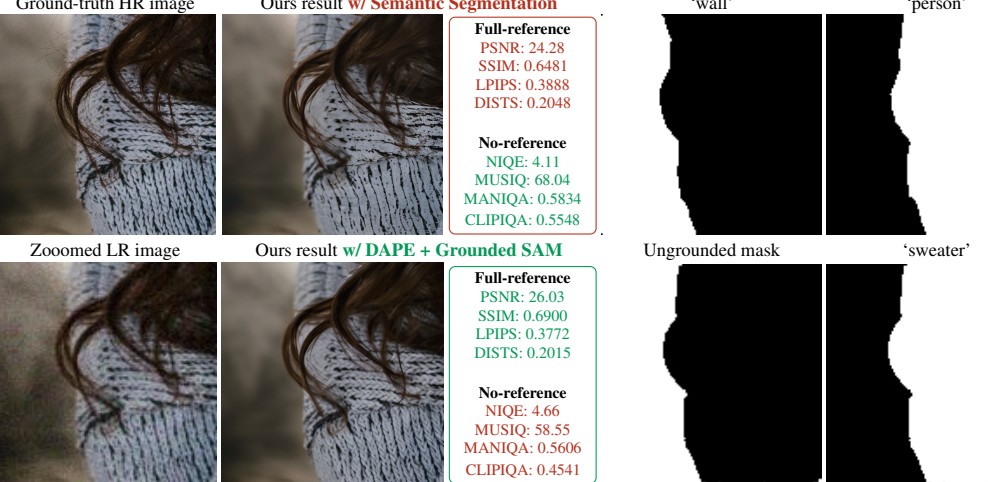

Figure S14: Column 1: Paired LR-HR images. Column 2: Qualitative comparisons of our results using semantic segmentation versus DAPE + Grounded SAM for prompt extraction and grounding. Column 3: Corresponding quantitative metrics. Columns 4–5: Grounded masks for selected tokens, where white indicates grounded regions. Due to the limited vocabulary of semantic segmentation models, the extracted prompt ('person') is a coarse descriptor and leads to imprecise restorations of the sweater's textures. In contrast, our method leverages DAPE with a broader vocabulary and grounds the more appropriate tag 'sweater' via Grounded SAM, resulting in semantically faithful restorations.

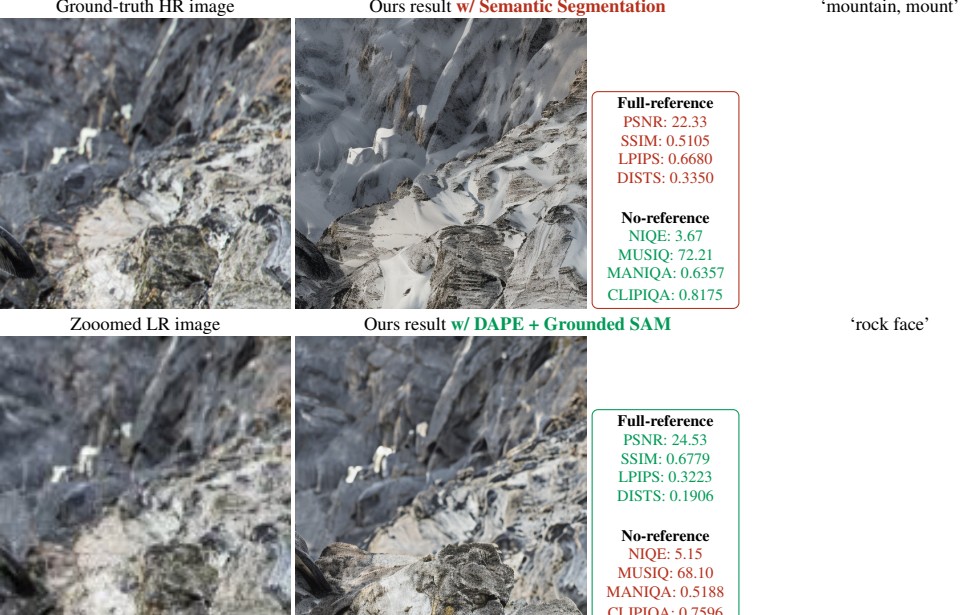

Figure S15: Column 1: Paired LR-HR images. Column 2: Qualitative comparisons of our results using semantic segmentation versus DAPE + Grounded SAM for prompt extraction and grounding. Column 3: Corresponding quantitative metrics. Columns 4: Grounded masks for selected tokens, where white indicates grounded regions. Both approaches ground the entire image with a single tag, as reflected by the fully white masks. However, due to the limited vocabulary of semantic segmentation models, the extracted prompt ('mountain, mount') is a coarse descriptor and leads to mountain-like hallucinations. In contrast, our method leverages DAPE with a broader vocabulary and grounds the more appropriate tag 'rock face' via Grounded SAM, resulting in semantically faithful restorations.

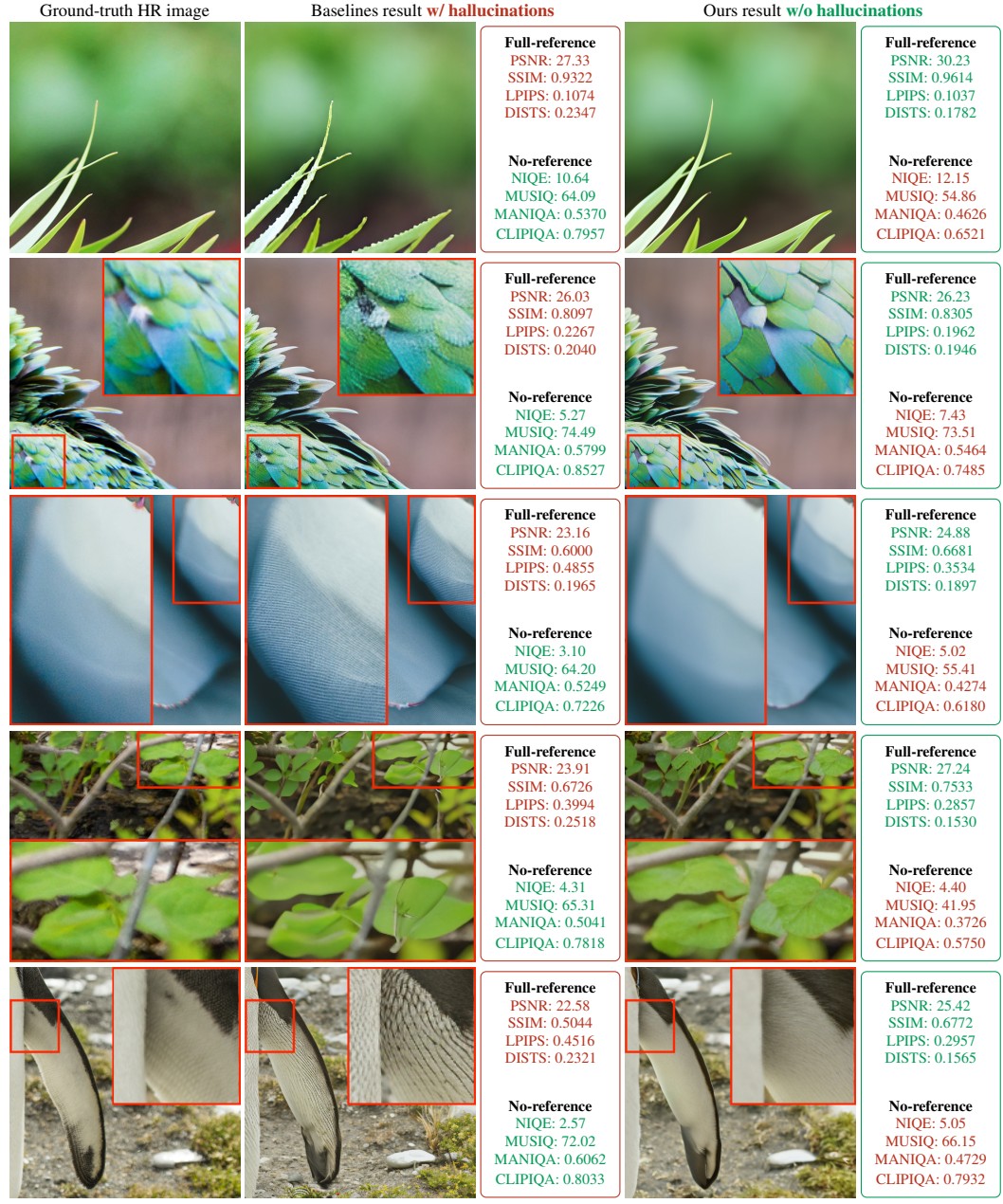

Figure S16: Additional qualitative results paired with quantitative metrics reveal that *no-reference metrics tend to misjudge and heavily reward hallucinated results*. This exposes their limitations in assessing semantic fidelity in super-resolution tasks.

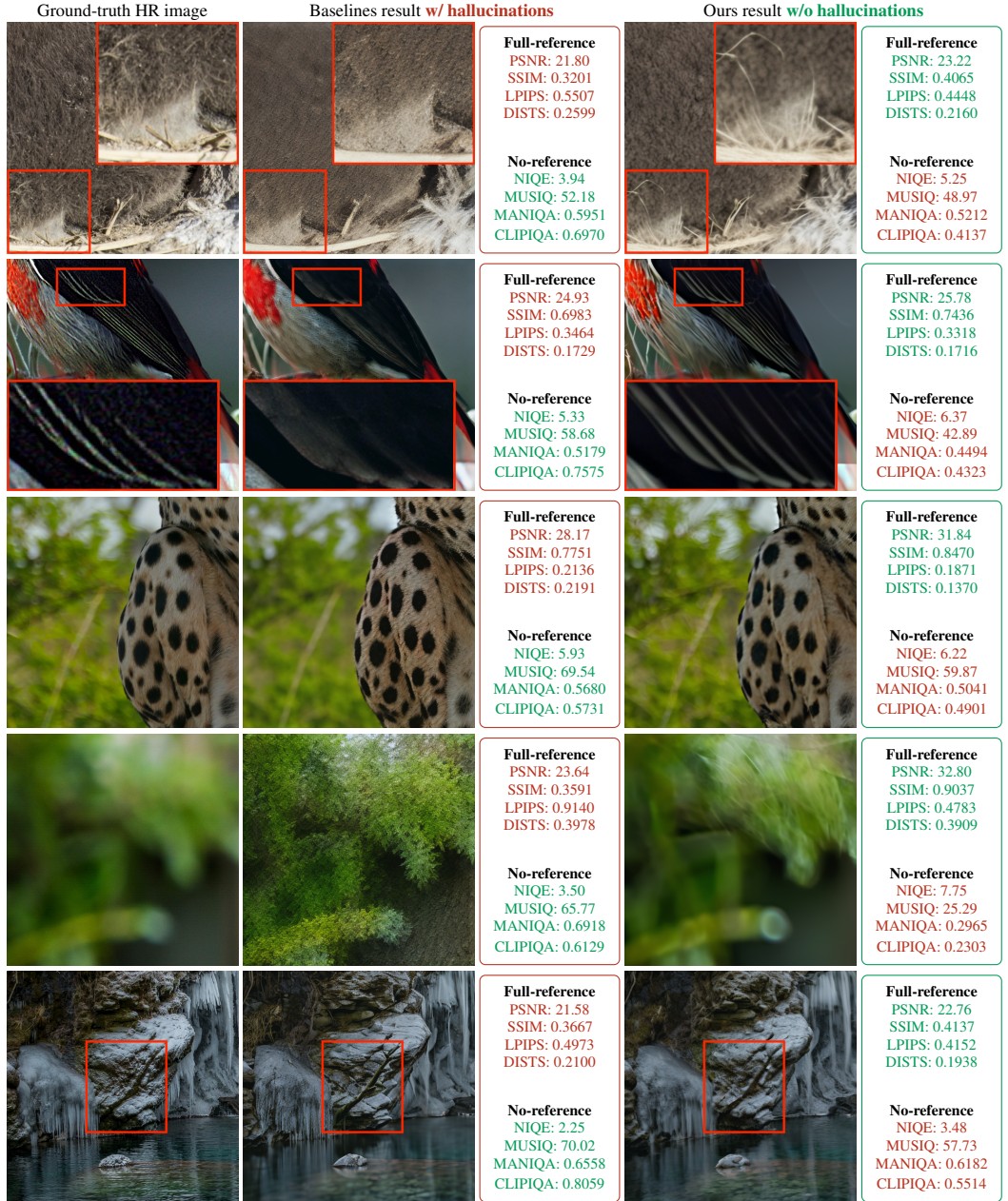

Figure S17: Additional qualitative results paired with quantitative metrics reveal that *no-reference metrics tend to misjudge and heavily reward hallucinated results*. This exposes their limitations in assessing semantic fidelity in super-resolution tasks.

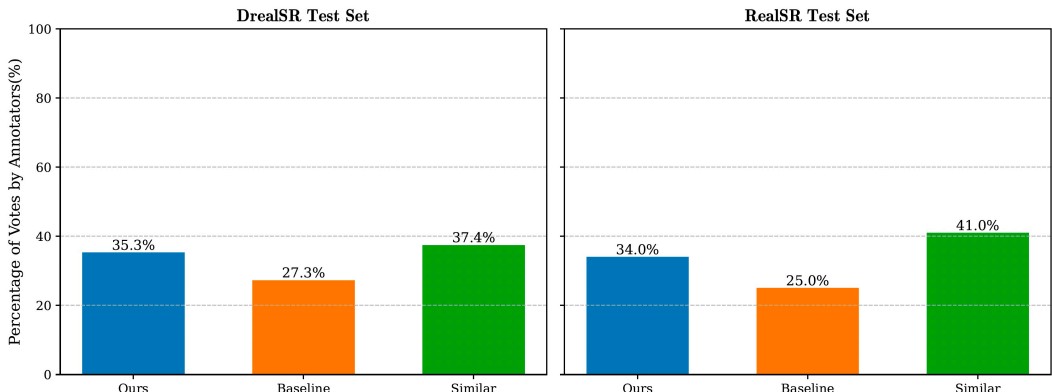

Figure S18: User study results evaluating human perceptual preferences between the baseline results and those produced after integrating SRSR. The study shows that human annotators consistently prefer our SRSR-enhanced outputs.

