# OpenReview forum: "SRSR: Enhancing Semantic Accuracy in Real-World Image Super-Resolution with Spatially Re-Focused Text-Conditioning"
_NeurIPS.cc/2025/Conference — NeurIPS 2025 poster_

### Official Review · Reviewer_pFBn · 2025-06-12

**Clarity:** 3
**Significance:** 3
**Originality:** 4
**Rating:** 5
**Confidence:** 5

**Summary:**

The paper proposes a novel Spatially Re-focused Super-Resolution (SRSR) framework for enhancing semantic accuracy in text-conditioned, diffusion-based real-world image super-resolution. The key idea is to refine text conditioning at inference time using visually grounded segmentation masks, which guide cross-attention (SRCA) and selectively disable classifier-free guidance (STCFG) for ungrounded regions. The approach is fully inference-time and plug-and-play, requiring no additional training. It outperforms seven state-of-the-art baselines in fidelity and perceptual metrics on both synthetic and real-world datasets.

**Questions:**

1. How does the performance change if the prompt extractor (DAPE) is replaced with a newer vision-language model or fine-tuned for specific domains?
2. Could the authors clarify how sensitive SRCA and STCFG are to errors in the segmentation masks from Grounded SAM?
3. Have the authors considered joint optimization between segmentation and SR to resolve interdependencies more robustly?
Score could increase with answers addressing generalization and robustness to segmentation or prompt extraction failures.

**Ethical Concerns:**

["NO or VERY MINOR ethics concerns only"]

**Final Justification:**

Thank you for the author's detailed and well-organized response. After reading it carefully, my response is as follows:
Regarding the shortcomings of DAPE, the author carefully analyzes the two improvements to SRCA he proposed to illustrate its robustness. This analysis is undoubtedly convincing. However, I was curious about what would happen if DAPE were replaced with other visual text models, and the author did not provide a good answer to this question.
For Q2, the author's reply gave me a better understanding of the robustness of the proposed STCFG and SAM.
For Q3, I would like to thank the author for recognizing the joint optimization strategy I proposed and listing it as a possible future work direction. I agree with the author that the performance of the proposed method will be better with the development of backbones.

**Limitations:**

Yes.

**Paper Formatting Concerns:**

No Obvious Formatting Concerns

**Quality:**

3

**Strengths And Weaknesses:**

Strengths:
1.High quality of methodology and implementation, with strong empirical evidence supporting the claims.
2.Clear ablation studies, hyperparameter analysis, and comparisons to state-of-the-art methods. Sufficient and reasonable experiments. Fig. 4 not only shows the superiority of the method, but also shows the limitations of existing IQAs.
3.Novel spatially-aware cross-attention and CFG mechanisms offer a unique contribution.
4.Strong plug-and-play usability makes it practical for integration.
Weaknesses:
1.Potential over-reliance on DAPE and Grounded SAM performance, which could vary in real-world applications.
2.No user study or real-world deployment evaluation to validate perceptual benefits from an end-user perspective. Fig. 4 displays the limitations of existing IQAs, which further indicates the necessity of user study to evaluate the motivation.

---

> ### Author Rebuttal · Authors · 2025-07-31
>
> We sincerely appreciate the reviewer’s thoughtful evaluation and high-confidence endorsement of our work regarding several important aspects:
>
> - **Novelty and originality**: Recognized in the summary (“novel SRSR framework”), the strengths (“novel SRCA and STCFG mechanisms offer a unique contribution,” “high quality of methodology and implementation”), and the numerical rating (“Originality: 4: excellent”).
> - **Comprehensive and compelling experimental results that validate effectiveness and identified research gaps**: Praised for “strong empirical evidence supporting the claims,” “clear ablation studies, hyperparameter analysis, and comparisons to state-of-the-art methods,” “outperforms seven state-of-the-art baselines in fidelity and perceptual metrics on both synthetic and real-world datasets,” “sufficient and reasonable experiments,” and “Fig. 4 not only shows the superiority of the method, but also shows the limitations of existing IQAs.”
> - **Practical impact**: Highlighted by “strong plug-and-play usability makes it practical for integration” and “the approach is fully inference-time and plug-and-play, requiring no additional training.”
>
> Finally, we value the reviewer’s constructive feedback in the weaknesses and questions sections, and we have addressed each point thoroughly in the responses below.
>
> ## W1, Q1, Q2, Q3: Generalization and robustness to segmentation and prompt extraction failures
>
> > Q3-2: “Score could increase with answers addressing generalization and robustness to segmentation or prompt extraction failures.”
> >
>
> We are grateful to the reviewer for explicitly outlining actionable criteria (Q3-2) that could lead to a higher rating, which is helpful and actionable for us to prepare a targeted response that can more effectively address the concern. Noting that several comments (W1, Q1, Q2, and Q3) closely relate to this point, we address them together for a cohesive and structured response.
>
> We first clarify both intuitively and with evidence, how our proposed SRCA and STCFG are robust to errors in prompt extraction (DAPE) and segmentation masks (Grounded SAM), thereby demonstrating that our framework does not overly rely on these components (addressing W1 and Q2). We then discuss potential future directions for improving these backbones, either by replacing DAPE (Q1) or by jointly optimizing the prompt extraction and segmentation (Q3-1).
>
> > W1: “Potential over-reliance on DAPE and Grounded SAM performance, which could vary in real-world applications.”
> >
>
> > Q2: “Could the authors clarify how sensitive SRCA and STCFG are to errors in the segmentation masks from Grounded SAM?”
> >
>
> ### (1) Robustness to failures of DAPE
>
> For reference, the three research gaps we identify in this work (Sec. 1, lines 34-46 and Sec. 3.1, lines 109-131) are:
>
> - G1: Diverted cross-attention
> - G2: Incorrect prompts being extracted
> - G3: Incomplete prompts and coverage (relevant prompts not being extracted)
>
> G2 and G3, both stemming from DAPE’s failures, often result in hallucinations in baseline methods. Our method is specifically designed to address these, thus being robust to them:
>
> - **Robustness to G2:** SRCA only retains tags that can be grounded and forms tag–mask pairs for attention re-focusing, thereby filtering out irrelevant tags wrongly extracted by DAPE. For example, in Fig. 2 (last two rows), DAPE extracts the irrelevant tag “camouflage,” causing hallucinations in the baseline, while our method removes its influence, as highlighted in the red box. Similar cases are presented in Supplementary Figs. S2 (”gun” and “rifle”) and S3 (”nose” and “mouse”).
> - **Robustness to G3:** Without SRCA, attention from existing tags can leak into ungrounded regions (e.g., backgrounds or salient objects missed by DAPE), resulting in hallucinated content. SRCA helps suppress this, but on its own cannot prevent global prompt tokens (e.g., EOS that captures the semantics of the entire prompt, consisting of all tags) from injecting semantics into ungrounded regions. STCFG further addresses this by blocking all text-conditioning (including EOS) in ungrounded regions. An illustrative failure case is shown in Fig. S1, where DAPE fails to extract the “claw” tag (highlighted in the red bounding box), leaving the region ungrounded. In this scenario, the baseline model (without SRCA and STCFG) hallucinates this area using unrelated tags such as “animal.” In contrast, our method (with SRCA and STCFG) prevents the attention of “animals” from influencing this region, resulting in a more faithful restoration. Similar examples can be seen in Fig. S7: when DAPE misses the “plate” tag, the corresponding region remains ungrounded. Baseline approaches allow irrelevant tags to introduce artifacts in such areas, while our framework effectively mitigates these errors and preserves semantic accuracy.
>
> ### (2) Robustness to failures of Grounded SAM
>
> We also observe failure cases where Grounded SAM does not provide comprehensive masks. Our method remains robust due to the design of STCFG, which improves faithfulness in ungrounded regions. Even if coverage is partial, SRCA ensures higher fidelity in grounded areas compared to the baseline. A visual example is shown in Fig. S1, where Grounded SAM fails to cover both “stone” objects in the image fully, and only the lower stone is confidently grounded, leaving a substantial ungrounded region. In the grounded area (yellow box), our method achieves higher fidelity. While the baseline hallucinates small pebble-like textures, our use of SRCA ensures that only the tag “animal” (and not “stone”) influences restoration, yielding more accurate results. For the ungrounded stone regions, STCFG effectively prevents over-synthesized or artifact-prone textures that the baseline exhibits, resulting in reconstructions much closer to the ground-truth HR image. Similar examples can be seen in Fig. 1 (Left). When the bird’s body is not fully grounded, STCFG prevents the introduction of artifacts from irrelevant tags such as “stone,” preserving the integrity of the ungrounded region.
>
> Additionally, our comprehensive hyperparameter analysis in Sec. 4.4 and Tab. 3 demonstrate that our method is robust to varying thresholds for Grounded SAM, consistently outperforming the baseline.
>
> ### (3) Potential directions
>
> > Q1: How does the performance change if the prompt extractor (DAPE) is replaced with a newer vision-language model or fine-tuned for specific domains?
> >
>
> > Q3-1: Have the authors considered joint optimization between segmentation and SR to resolve interdependencies more robustly?
> >
>
> We appreciate these insightful suggestions. Following the baselines of SeeSR and OSEDiff, DAPE is adopted due to its proven advantage in Real-ISR in terms of degradation-awareness and efficiency, outperforming other off-the-shelf models like ResNet, YOLO, BLIP, and LLaVA. This is because DAPE is already fine-tuned on the Recognize Anything Model (RAM) for this domain, as suggested in Q1.
>
> Nevertheless, as identified in G2 and G3 and previously discussed, there remains room for improvement. As discussed in the future work section (Supplementary Sec. E), we believe that advances in the prompt extraction and segmentation backbones can further improve SRSR’s performance. This is grounded on our observation from ablation studies that stronger semantic segmentation backbones (e.g., DINO-X over Mask2Former) lead to better results. Also, in line with Q3-1, we believe joint optimization of segmentation and SR is a promising and novel direction that could further enhance the current backbones. We are grateful that the reviewer endorses our comprehensive experiments and compelling results, which show that our current DAPE + Grounded SAM backbones already achieve consistent state-of-the-art results. As new advances emerge in these backbones, we believe our method will benefit from orthogonal improvements.
>
> ## W2: User study
>
> > No user study or real-world deployment evaluation to validate perceptual benefits from an end-user perspective. Fig. 4 displays the limitations of existing IQAs, which further indicates the necessity of user study to evaluate the motivation.
> >
>
> Thank you for highlighting the importance of evaluating perceptual benefits from an end-user perspective. In response to this motivation, we have conducted a user study to evaluate human perceptual preferences between the baseline results and those produced after integrating SRSR, details of which are included in Supplementary Material Sec. D and Fig. S18. In our study, we presented the low-resolution input for each image along with two anonymized high-resolution outputs (ours: SRSR-SeeSR and the baseline: SeeSR) labeled ‘Method A’ and ‘Method B’. Annotators were asked to select the preferred image or indicate if both appeared visually similar, with evaluations based on sharpness, visual realism, and detail preservation. The order of presentation was randomized, and full images were shown (not crops) to avoid bias. Results are aggregated and visualized in bar charts (Figure S18), showing that SRSR-SeeSR is consistently preferred over the baseline SeeSR across both datasets.

---

### Official Review · Reviewer_u37U · 2025-06-28

**Clarity:** 4
**Significance:** 3
**Originality:** 3
**Rating:** 5
**Confidence:** 5

**Summary:**

This paper introduces SRSR, a plug-in framework designed to improve semantic accuracy in real-world image super-resolution (Real-ISR). It features two key components: (1) Spatially Re-focused Cross-Attention (SRCA), which restricts cross-attention to semantically grounded regions defined by tag–mask pairs, and (2) STCFG, which mitigates hallucinations in ungrounded areas by adaptively modulating classifier-free guidance. SRSR is compatible with existing models without retraining. Experiments on demonstrate consistent improvements in image fidelity.

**Questions:**

**Motivation clarity**
1. In Lines 37–38, you mention that DAPE “is only partially robust to severe degradations.” Could you elaborate on how this conclusion was derived? Is it based on specific failure cases, empirical observations, or quantitative results?

**Perception–fidelity trade-off**

2. In Fig. 7 of SUPIR$^{[1]}$, qualitative examples suggest that non-reference scores better reflect visual quality. In contrast, your work implies that improvements in reference-based metrics (e.g., PSNR, SSIM) better correlate with perceptual quality, and that gains in non-reference metrics may actually result from hallucinations. While I understand that this trade-off remains a challenging and open question in Real-ISR, and I appreciate the qualitative evidence you provide, there appears to be a contradiction with SUPIR’s conclusions. Could you offer further insight into this discrepancy? Specifically, do you see a path toward a more generalizable understanding of the perception–fidelity trade-off, beyond the current reliance on metric scores and qualitative examples?

**Source of improvement: SRCA vs STCFG**

3. In Section B, you show that replacing DAPE with semantic segmentation degrades performance in your framework. Could this be due to segmentation-based method has ungrounded regions where CFG can be selectively reduced? As noted in the weaknesses section, the marginal performance gap between V4 and V6 in Table 2 suggests that most of the gains may stem from CFG reduction (via STCFG), rather than from SRCA. Also, the segmentation-based variant yields better non-reference scores, though you suggest these may prefer artifact-heavy outputs. To clarify this: (1) A direct comparison between **(a) DAPE + Grounded SAM** and **(b) semantic segmentation, both with and without CFG reduction**, would help isolate the individual impact of each component; and
(2) Further insight on the perception–fidelity trade-off (as discussed in Question 2) could support this claim.

**Incorporating SRSR into training**

4. In Section E, you suggest that integrating SRSR into the training process could lead to further improvements. Could you provide more intuition or preliminary evidence to support this claim? For example, would incorporating grounded attention during training help the model learn better semantic localization?

[1] Yu *et al.* "Scaling Up to Excellence: Practicing Model Scaling for Photo-Realistic Image Restoration In the Wild"

**Ethical Concerns:**

["NO or VERY MINOR ethics concerns only"]

**Final Justification:**

The authors' responses have addressed all of my concerns. The paper is insightful, well-written, and technically sound, contributing to the field of Real-ISR. Therefore, I would like to raise my score to 5 and suggest that this paper be accepted.

**Limitations:**

I would expect the authors to provide a deeper explanation for the significant drop in no-reference metrics, which are currently regarded as important indicators of performance in SD-based Real-ISR frameworks. This issue raises concerns about the practicality of the proposed method in real-world applications, where reference (ground truth) images are typically of low quality and no-reference metrics play a crucial role in evaluating perceptual quality.

**Paper Formatting Concerns:**

No.

**Quality:**

3

**Strengths And Weaknesses:**

**Strength**

1. The paper is well-motivated, clearly identifying critical and unsolved issues in SD-based Real-ISR frameworks. The illustrative examples and experiments tightly support the claims.
2. This paper is technically sound and clearly written.
3. The proposed solutions are elegant:
    - SRCA restricts cross-attention to semantically grounded regions (identified via Grounded-SAM), reducing the influence of irrelevant tags.
    - STCFG introduces a novel use of classifier-free guidance in Real-ISR. Instead of manually tuning a global parameter, it disables CFG in ungrounded regions, reducing hallucinations.
4. The plug-in framework enhances Real-ISR models without the need of retraining, making it efficient and practical.
5. The experiments are comprehensive, and the qualitative results are particularly convincing.

**Weaknesses**

1. The non-reference scores drop significantly when SRSR is applied, raising the concern that the improvements may largely stem from simply reducing classifier-free guidance (CFG) through the proposed STCFG, rather than targeted semantic enhancement.
    - For example, the ablation study (Table 2) shows that while STCFG substantially improves fidelity (from V2 to V6), the performance gaps between V6 and V4, as well as between V3 and V4, are marginal. This suggests that the gains may primarily result from reducing CFG strength in ungrounded regions, rather than from the semantic refinement provided by SRCA.
    - To clarify this point, could the authors report the no-reference metric scores for the V2, V3, and V4 models?

2. The grounding threshold described in Section 4.4 appears to act as another hyper-parameter that influences the perception–fidelity trade-off. While removing redundant tags is a reasonable strategy, its impact on performance is not clearly reflected in the results (e.g., the improvement from V1 to V2 is small).

---

> ### Author Rebuttal · Authors · 2025-07-31
>
> We sincerely thank the reviewer for their detailed evaluation and positive assessment of our work. We are especially encouraged by the reviewer’s recognition of our paper’s strengths across all four criteria: quality, clarity, originality, and significance:
>
> - **Motivation and identified research gaps**: “The paper is well-motivated, clearly identifying critical and unsolved issues in SD-based Real-ISR frameworks.”
> - **Novelty and methodological elegance**: “The proposed solutions are elegant,” and “STCFG introduces a novel use of classifier-free guidance in Real-ISR.”
> - **Comprehensive and convincing experiments**: “The experiments are comprehensive, and the qualitative results are particularly convincing,” “Experiments demonstrate consistent improvements in image fidelity,” and “The illustrative examples and experiments tightly support the claims.”
> - **Efficiency and practical merits**: “The plug-in framework enhances Real-ISR models without the need of retraining, making it efficient and practical,” and “SRSR is compatible with existing models without retraining.”
> - **Clarity and technical soundness**: Recognized by the numerical rating (“Clarity: 4: excellent”) and the encouraging comment: “This paper is technically sound and clearly written.”
>
> Finally, we appreciate the reviewer’s constructive feedback in the weaknesses and questions sections, and we address each point thoroughly in the responses below.
>
> ## W1, Q2, Q3, and Limitation: Source of improvement and limitation of no-reference metrics
>
> Given the close connections among W1, Q2, Q3, and the Limitation section, we address them together to provide a cohesive and structured response. We first disentangle the source of improvement between SRCA and STCFG (addressing W1 and Q3), and then offer further insights regarding the limitations of current no-reference metrics and the broader perception–fidelity trade-off (addressing Q2 and Limitation, which are also relevant to W1 and Q3).
>
> ### **(1) Source of improvement: SRCA vs STCFG**
>
> > W1: Improvements may largely stem from simply reducing classifier-free guidance (CFG) through the proposed STCFG, rather than targeted semantic enhancement.
> >
>
> > Q3: Source of improvement: SRCA vs STCFG.
> >
>
> We appreciate the reviewer’s thoughtful comments and agree that STCFG is the main driver of performance gains. However, we have noticed a slight but crucial misconception in W1 comments; thus, it is important to clarify that **the improvements from STCFG are not independent of targeted semantic enhancement—rather, both SRCA and STCFG contribute synergistically to this goal** (instead of SRCA solely contributing towards this). The reason why STCFG is more significant, in a nutshell, is that it effectively addresses the critical and widely observed shortcomings of grounding and SRCA (w/o STCFG) that are otherwise detrimental to semantic accuracy. Thus, it is important to highlight that such improvements are also for semantic accuracy. We will elaborate in the following paragraphs.
>
> SRCA refines semantic conditioning via more precise cross-attention maps, but in ungrounded areas, its effect in ungrounded areas is limited by the persistence of summary tokens (e.g., EOS) that capture semantics of all tags in the prompt (including the mismatched tags) and meaningless separator tokens, which can inadvertently introduce mismatched or noisy semantics. Note that masking these tokens is not feasible, as it disrupts the diffusion model’s noise prediction.
>
> Moreover, grounding (required for SRCA) reduces prompt coverage and increases ungrounded regions, especially for degraded LR images; thus, although SRCA alone has clear benefits, substantial room for improvement remains.
>
> STCFG addresses the limitations of SRCA by replacing text-conditioned noise with unconditional noise in ungrounded regions, reducing hallucinations while preserving SRCA’s benefits.
>
> Lastly, regarding the comparisons with segmentation-based methods in our ablation studies (V7 and V8 in Tab. 2), we wish to clarify that both versions already incorporate STCFG to ensure fair comparison with our final choice. Removing STCFG (see additional results: V9, V10) further degrades performance, highlighting its importance. With semantic segmentation (SS), the increased coverage reduces the frequency of ungrounded regions, so STCFG is less often triggered and brings smaller additional benefit than the setting without SS.
>
> | Version | PSNR↑ | SSIM↑ | LPIPS↓ | DISTS↓ |
> | --- | --- | --- | --- | --- |
> | V7 | **26.3128** | **0.7620** | **0.2725** | **0.2093** |
> | V9 | 26.2885 | 0.7609 | 0.2734 | 0.2098 |
> |  |  |  |  |  |
> | V8 | **26.3449** | **0.7627** | **0.2722** | **0.2089** |
> | V10 | 26.3221 | 0.7621 | 0.2729 | 0.2090 |
> |  |  |  |  |  |
>
> ### **(2) Limitation of no-reference metrics and the Perceptual-Fidelity trade-off**
>
> > Q2: Perception–fidelity trade-off.
> >
>
> > Limitation: No-reference metrics.
> >
>
> We appreciate the reviewer’s acknowledgment that this remains a challenging and open question in Real-ISR. The drop in no-reference metric scores is due to the same improvements in semantic accuracy (i.e., reduced hallucinations). No-reference metrics tend to over-reward artifact-heavy or hallucinated outputs, as noted in our paper and acknowledged by the reviewer. In contrast, full-reference metrics we employed also have dedicated measures for evaluating perceptual quality (LPIPS and DISTS) and fidelity (PSNR and SSIM). Our improvements in these metrics are well-aligned with the visual evidence we provide in the paper, such as Fig. 4, and Supplementary Material Figures S1-S17.
>
> In SUPIR (Fig. 7), human preference can favor creative outputs diverging from the reference, but in Real-ISR, especially for applications needing semantic faithfulness (e.g., e-commerce product images), hallucinations reduce trust and utility. Thus, while SD-based SR models excel in producing creative results that are visually pleasing, an important question is that shall we reward creativity in Real-ISR as much as possible by pursuing favorable no-reference metrics results? Our view is that **faithful restoration, as measured by full-reference fidelity and perceptual quality metrics, should remain a central goal in many practical contexts**. For V2–V4, we focus on full-reference metrics: V3 (SRCA only) improves fidelity but reduces perceptual quality, while V4 (SRCA+STCFG) significantly improves both, aligning with qualitative evidence.
>
> **Path forward**: We fully agree that in scenarios lacking references, reliance on existing no-reference metrics is problematic, and developing improved, hallucination-aware no-reference metrics that better align with full-reference scores and human perception is an urgent research need. Promising directions include vision-language model-based assessment and metrics penalizing hallucinations. Nonetheless, in the standard settings of Real-ISR and similar settings when references are available, we advocate for full-reference metrics as a more complete and reliable evaluation. We appreciate the reviewer’s feedback and will incorporate these points in our revised Limitations section.
>
> ## W2: The contribution of grounding
>
> Thanks for raising this point. While grounding improves prompt accuracy by removing redundant tags, it also reduces coverage, increasing ungrounded regions. Thus, the modest gain from V1 to V2 reflects a suboptimal trade-off. Our STCFG module compensates for this limitation by improving quality in ungrounded regions. As a result, SRSR is robust to grounding thresholds (Tab. 3), consistently significantly outperforming the baseline.
>
> We also wish to emphasize that grounding's contribution extends significantly beyond the performance improvements from V1 to V2. Crucially, it provides the essential tag–mask pairs necessary for enabling our SRCA and STCFG modules, without which these key innovations would not be feasible.
>
> ## Q1: Motivation clarity
>
> Thank you for this question. Our conclusion that “DAPE is only partially robust to severe degradations” is based on failure case analysis, empirical observations, cross-attention map analysis, and quantitative comparisons.
>
> For reference, our study identifies three main research gaps (Sec. 1, lines 34–46 and Sec. 3.1, lines 109–131):
>
> - G1: Diverted cross-attention
> - G2: Incorrect prompts being extracted
> - G3: Incomplete prompts and coverage (relevant prompts not being extracted)
>
> We observed hallucinated objects in the baseline linked to DAPE-extracted tags not grounded in the image (G2). Cross-attention maps confirmed their impact. We also found that DAPE sometimes misses relevant tags, leading to ungrounded regions and further hallucinations (G3). This is supported by qualitative and quantitative evidence throughout the paper and the supplementary materials. For example:
>
> - **G2 (Incorrect tags):** In Fig. 2 (last two rows), DAPE extracts the irrelevant tag “camouflage,” leading to hallucinations in the baseline. Similar cases appear in Supplementary Figs. S2 (“gun” and “rifle”) and S3 (“nose” and “mouth”), etc.
> - **G3 (Incomplete tags):** In Fig. S1, DAPE fails to extract the “claw” tag, resulting in an ungrounded region where the baseline hallucinates this area based on unrelated tags such as “animal.” A similar pattern occurs in Fig. S7, where missing the “plate” tag allows irrelevant tags to introduce artifacts.
>
> ## Q4: Incorporating SRSR into training
>
> We appreciate the reviewer’s interest in training-time integration of SRSR, which we briefly mentioned in our Future Works section. Our intuition is that incorporating grounding, SRCA, and STCFG during training would align the model’s learning with the spatial and semantic constraints imposed at inference, improving region-specific supervision and semantic localization. Integrating STCFG in training would also improve robustness to incomplete or noisy tags. We consider this a promising direction for future work.

---

> > ### Comment · Reviewer_u37U · 2025-08-06
> >
> > Dear Authors,
> >
> > Thank you for the detailed reply, which addresses most of my concerns. I especially appreciate your thoughts on the quality assessment metrics for Q2.
> >
> > I still have a couple of follow-up points:
> >
> > 1. Your rebuttal explains that "STCFG addresses the limitations of SRCA by replacing text-conditioned noise with unconditional noise in ungrounded regions." However, my understanding is that STCFG decreases classifier-free guidance in ungrounded regions, as mentioned in Lines 51-54 that STCFG disables classifier-free guidance in ungrounded regions. Could you please clarify this point?
> >
> > 2. I was also looking for the results on no-reference metrics in the table in your rebuttal. I believe adding those scores and a brief discussion would make your points even stronger.
> >
> > 3. I agree that full-reference metrics are better for judging perceptual quality. However, since the ground truth images in datasets like RealSR/DRealSR are of low quality, the results can be hard to interpret. I think your view would be further supported if higher-quality test sets are available.
> >
> > If you can add the full results for the no-reference metrics to the table, along with a discussion, I would be happy to raise my score.
> >
> > Thank you again!

---

> > > ### Author Response · Authors · 2025-08-07
> > > **Response to follow-up Q2 and Q3**
> > >
> > > > Follow-up Q2: Results on no-reference metrics and discussions
> > > >
> > >
> > > Thank you for this valuable suggestion. We fully agree that including the no-reference metrics in the ablation table can further strengthen our point that the existing no-reference metrics are flawed, as they tend to over-reward artifact-heavy or hallucinated outputs.
> > >
> > > For the main comparisons with baselines (Tab. 1), we have already provided the full results that contain both full-reference and no-reference metrics. In addition, as you kindly suggested, we now provide the full results for the ablation study (Tab. 2), including all no-reference metrics. We also include the results for the extra versions (V9 and V10) according to the valuable suggestions in your initial review, which further enriches the comprehensiveness of the ablation analysis.
> > >
> > > | Version | PSNR↑ | SSIM↑ | LPIPS↓ | DISTS↓ | FID↓  | NIQE↓  | MUSIQ↑  | MANIQA↑  | CLIPIQA↑ |
> > > | --- | --- | --- | --- | --- | --- | --- | --- | --- | --- |
> > > | V1: SeeSR | 25.1717 | 0.7219 | 0.3008 | 0.2223 | 125.55 | 5.4081 | 69.77 | 0.6442 | 0.6612 |
> > > | V2: SeeSR + Grounding | 25.1751 | 0.7234 | 0.3001 | 0.2229 | 128.41 | 5.4852 | 69.84 | 0.6436 | 0.6701 |
> > > | V3: SeeSR + Grounding + SRCA | 25.2688 | 0.7280 | 0.3013 | 0.2254 | 132.55 | 5.5029 | 69.91 | 0.6422 | 0.6683 |
> > > | V4: SeeSR + Grounding + SRCA + STCFG (Ours) | 26.3996 | 0.7632 | 0.2718 | 0.2092 | 126.31 | 5.8627 | 62.88 | 0.5628 | 0.5409 |
> > > | V5: SeeSR + Grounding + SRCA + STCFG + Ungrounded Tags | 26.3871 | 0.7625 | 0.2729 | 0.2095 | 126.28 | 5.8652 | 63.15 | 0.5665 | 0.5460 |
> > > | V6: SeeSR + Grounding + STCFG | 26.3986 | 0.7627 | 0.2735 | 0.2112 | 127.24 | 5.8943 | 62.99 | 0.5648 | 0.5456 |
> > > | V7: SeeSR + Grounding + SRCA + STCFG + Mask2Former SS | 26.3128 | 0.7620 | 0.2725 | 0.2093 | 127.51 | 5.8026 | 63.31 | 0.5689 | 0.5477 |
> > > | V8: SeeSR + Grounding + SRCA + STCFG + DINO-X SS | 26.3449 | 0.7627 | 0.2722 | 0.2089 | 127.77 | 5.8190 | 63.17 | 0.5670 | 0.5450 |
> > > | V9: SeeSR + Grounding + SRCA + Mask2Former SS | 26.2885 | 0.7609 | 0.2734 | 0.2098 | 127.19 | 5.7801 | 63.73 | 0.5728 | 0.5543 |
> > > | V10: SeeSR + Grounding + SRCA + DINO-X SS | 26.3221 | 0.7621 | 0.2729 | 0.2090 | 127.33 | 5.7927 | 63.58 | 0.5713 | 0.5520 |
> > >
> > > **Discussion:**
> > >
> > > This table highlights a consistent trend: improvements in full-reference fidelity metrics (PSNR, SSIM) and perceptual quality metrics (LPIPS, DISTS) align with observed improvements in visual and semantic fidelity, as supported by our qualitative analysis (e.g., Fig. 4, Supplementary Figs. S1–S17). However, the same changes often lead to *worse* results in all no-reference metrics (NIQE, MUSIQ, MANIQA, CLIPIQA), which can misleadingly favor outputs with hallucinated or artifact-heavy details.
> > >
> > > - For example, removing STCFG (V3) from our full version (V4) improves all no-reference metrics, even though fidelity, full-reference perceptual quality, and visual quality drop.
> > > - Similarly, removing SRCA (V6) worsens both fidelity and full-reference perceptual quality, yet again, no-reference metrics do not consistently reflect this decline.
> > > - This pattern also holds when comparing DINO-X to Mask2Former (V8/V7, V10/V9): fidelity, full-reference perceptual quality metrics, and visual quality are better with DINO-X, which aligns with the inherent advantages of DINO-X over Mask2Former, as we elaborated in the paper. However, no-reference metrics show the reverse trend.
> > >
> > > These findings reinforce our point in the rebuttal Q2: current no-reference metrics often fail to penalize hallucinations and can mislead practical evaluation, especially in Real-ISR. We will include this comprehensive table and its discussion in our revision. Thank you again for your thoughtful recommendation and for helping us further strengthen this point.
> > >
> > > > Follow-up Q3: Higher-quality test sets can further support the advantages of full-reference metrics for judging perceptual quality.
> > > >
> > >
> > > Thank you for this insightful comment and for sharing your agreement with our view that full-reference metrics are better for judging perceptual quality. In this work, we have followed the prevailing standards in the Real-ISR literature, evaluating our approach on the DIV2K-Val test set in addition to RealSR and DRealSR, and show consistent improvements. We fully agree that as higher-quality test sets become available, they will further strengthen the case for using full-reference metrics as the most trustworthy indicators of perceptual quality in Real-ISR. We appreciate your insightful perspective and will highlight this point in our revision.
> > >
> > > Thank you once again for your detailed and insightful feedback and your encouraging support. Please let us know if any further clarifications are needed, and we'd be happy to provide them.

---

> > > > ### Comment · Reviewer_u37U · 2025-08-09
> > > >
> > > > Dear Authors,
> > > >
> > > > Thank you for the thorough clarification, which has addressed all of my concerns.
> > > >
> > > > I find your argument for using full-reference metrics as an evaluation standard highly compelling. Standardizing evaluation is a critical issue for the field (e.g., whether to fix the classifier-free guidance magnitude, the manually added prompts, ...), and your paper makes a nice contribution toward this goal. This will not only enable more reliable comparisons but also help drive innovation in creating high-quality datasets.
> > > >
> > > > In light of your responses, I have raised my score and recommend this paper to be accepted.
> > > >
> > > > Best regards,

---

> > > > > ### Author Response · Authors · 2025-08-09
> > > > >
> > > > > Thank you very much for letting us know that our responses have addressed all of your concerns. We are especially grateful for your recognition that our argument for using full-reference metrics is highly compelling and that our work makes a nice contribution towards the goal of addressing a critical issue in the field, enabling more reliable comparisons and helping to drive future innovations. We also sincerely appreciate your consistent positive recommendation of this work and your decision to further raise your score in light of our responses. We will incorporate the points discussed in our exchanges, inspired by your valuable suggestions, into the revised version of the paper, as we believe they will indeed further strengthen our points and contributions, just as you kindly suggested.

---

> ### Author Response · Authors · 2025-08-07
> **Response to follow-up Q1**
>
> Thank you very much for your careful review of our rebuttal and for these detailed follow-up questions. We are grateful to hear that our rebuttal has successfully addressed your concerns, and our additional insights on the limitations of existing no-reference metrics and their path forward are helpful. We are happy to address your insightful follow-up questions comprehensively one by one, as follows:
>
> > Follow-up Q1: Clarification of “STCFG addresses the limitations of SRCA by replacing text-conditioned noise with unconditional noise in ungrounded regions.”
> >
>
> Thank you for raising this important point. We would like to clarify that the quoted explanation in our rebuttal is technically equivalent to “STCFG disables classifier-free guidance in ungrounded regions” (as stated in Lines 51–54 of the paper). The core idea is captured in our noise prediction equations, as detailed below.
>
> Formally, as shown in Eq. 6 of the paper, the noise prediction at each pixel $i$ is defined as:
>
> $$\hat{\epsilon}_i \leftarrow (1 - M_i)({\epsilon}_θ(x_t, \phi) + s ({\epsilon}_θ(x_t, y)-{\epsilon}_θ(x_t, \phi))) + M_i {\epsilon}_θ(x_t, \phi) \tag{6}$$
>
> where $M$ is the ungrounded mask for each image, and $M_i \in \{0,1\}$ indicates whether pixel $i$ is ungrounded.
>
> - **For grounded regions (**$M_i=0$**)**, the equation simplifies to the standard classifier-free guidance (CFG) formulation, where the noise prediction leverages both unconditional and text-conditional branches.
>
> $$\hat{\epsilon}_i \leftarrow {\epsilon}_θ (x_t, \phi) + s ({\epsilon}_θ(x_t, y) - {\epsilon}_θ (x_t, \phi)) \tag{6a}$$
>
> - **For ungrounded regions (**$M_i=1$**)**, the equation becomes only using the unconditional noise prediction, while the text-conditional term is fully removed:
>
> $$\hat{\epsilon}_i \leftarrow {\epsilon}_θ (x_t, \phi)  \tag{6b}$$
>
> Comparing these two forms, it is clear that for ungrounded regions (Eq. 6b):
>
> - STCFG *replaces* the text-conditioned noise $\epsilon_\theta(x_t, y)$ with unconditional noise $\epsilon_\theta(x_t, \phi)$ in Eq. 6a, resulting in the second term being eliminated rather than just decreasing the guidance strength, becoming Eq. 6b. This represents the explanation in our rebuttal that “STCFG addresses the limitations of SRCA by replacing text-conditioned noise with unconditional noise in ungrounded regions.”
> - This directly corresponds to “disabling classifier-free guidance” of Eq. 6a (as we mentioned in Lines 51-54), as the guidance (the difference between conditional and unconditional branches) is set to zero. As a result, Eq. 6b represents how diffusion models conduct unconditional generation (i.e., disabling the text-conditional generation capability provided by the second term of CFG).
>
> We hope this clarifies that both descriptions are fully consistent. We will ensure this equivalence is clearly stated in our revision. Thank you again for giving us the opportunity to clarify this technical point.

---

### Official Review · Reviewer_pH61 · 2025-06-30

**Clarity:** 3
**Significance:** 2
**Originality:** 3
**Rating:** 4
**Confidence:** 4

**Summary:**

This paper proposes a plug-and-play spatially re-focused cross-attention based on the discovery that traditional cross-attention tends to divert towards irrelevant pixels. Combined visual grounding, the devised super-resolution framework can effectively avoid the interference of unrelated pixels.

**Questions:**

Please refer to the weakness.

**Ethical Concerns:**

["NO or VERY MINOR ethics concerns only"]

**Final Justification:**

Most concerns have been addressed.

**Limitations:**

yes

**Paper Formatting Concerns:**

No.

**Quality:**

2

**Strengths And Weaknesses:**

**Strength**
1. Visually, this work reveals the fact that the vanilla cross-attention induces hallucinations, while the devised spatially re-focused cross-attention can address this issue in some extent.
2. The fidelity of super-resolved images are significantly improved.

**Weakness**
1. The design of the proposed modules are not novel, which are guided by the segmentation masks.
2. The writing should be further improved. For instance, the abstract is too short and the first paragraph is too long.
3. In Figure 1 (left), the body of bird is not highlighted in the attention matrix, could you explain the reason?
4. The references are not sufficient, e.g., SFT, CoSeR, etc.
5. As for the issues about uncoverage of full-image, why not consider employing the models for open word panoramic segmentation？
6. Authors are prefer to emphasize the strength of the designed modules before explanation, e.g., Sec. 3.2.
7. Typo. Wrong tense is used in line 179.
8. There are some repeated references, e.g., 9 and 10,45 and 46.

---

> ### Author Rebuttal · Authors · 2025-07-31
>
> We sincerely thank the reviewer for recognizing several important strengths of our work in the **Summary** section, including:
>
> - the **plug-and-play design** of our spatially re-focused cross-attention module,
> - our **novel discovery** that traditional (vanilla) cross-attention induces hallucinations by diverting attention toward irrelevant pixels, and
> - the **effectiveness of our strategy** in mitigating interference from unrelated pixels.
>
> We further appreciate the reviewer highlighting in the **Strengths** section that, based on our aforementioned **novel discovery/reveal** of a critical research gap, our proposed method can:
>
> - visibly **alleviate hallucinations**, and
> - significantly **improve the fidelity** of the super-resolved images.
>
> We also appreciate the valuable feedback listed in the **Weaknesses** section, and we address them thoroughly in the responses below.
>
> ## W1: Novelty
>
> We appreciate the reviewer for raising this critical point about novelty. We are encouraged that the reviewer’s **originality** rating is “3: good,” and that all other reviewers gave either “3: good” or “4: excellent,” with explicit remarks such as “novel SRCA and STCFG mechanisms offer a unique contribution,” “STCFG introduces a novel use of classifier-free guidance in Real-ISR,” and “novel SRSR framework.”
>
> We respectfully wish to clarify a key misunderstanding: our framework goes substantially beyond simply using segmentation masks as additional priors. Before elaborating on our specific novel contributions, we wish to provide a high-level summary that the novelty of our approach lies in **how we (1) design and (2) deploy the spatial prior**, addressing several under-explored and fundamental research gaps in text-conditioned Real-ISR.
>
> - **Design motivation**: Rather than using off-the-shelf segmentation masks as generic priors, our masks are constructed through *grounded* degradation-aware tags, and our entire framework is motivated by three systematically identified research gaps (Sec. 1, 3.1):
>     - **G1**: Diverted cross-attention (Novel finding that traditional cross-attention leads to hallucinations by diverting attention to irrelevant pixels.)
>     - **G2**: Incorrect prompts being extracted (Although many works attempt to develop better text extraction methods for Real-ISR, we identify that even state-of-the-art, degradation-aware prompt extractors can still frequently output incorrect tags due to heavily degraded LR inputs.)
>     - **G3**: Incomplete prompt coverage.
> - **Deployment**: Rather than simply adding segmentation as a conditioning signal, our modules **refine the core cross-attention and CFG mechanisms**, changing where and how the model focuses semantic information. This approach fundamentally differs from previous work and is specifically aimed at enhancing semantic accuracy, a largely under-explored challenge.
>
> ### Specific Novel Contributions
>
> - **SRCA** (Spatially Re-focused Cross-Attention) directly addresses G1 and G2:
>     - G1: To our knowledge, **no prior work has explicitly re-designed the cross-attention mechanism itself to resolve its inherent limitations**. Prior methods typically focus on improved text prompts or additional conditioning, while continuing to use default cross-attention. Our work is motivated by our novel observation that unrefined cross-attention causes semantic hallucinations—an insight backed by new empirical evidence (see ablation and visualizations).
>     - G2: Instead of similarly exploring better prompt extractors, **we are the first to use the idea of grounding in this task to improve prompt accuracy further**.
> - **STCFG** (Spatially Targeted Classifier-Free Guidance) addresses G3:
>     - Simply using segmentation masks to guide restoration (as the reviewer assumed to be our design) can improve coverage; we have actually conducted ablation studies (Tab. 2) to prove the inferiority of this design. We have gone several extra miles, such as analyzing the limitations of using SRCA alone and other ablated versions. These motivated us to design STCFG, a **novel refinement of CFG tailored to Real-ISR**. It dynamically disables CFG in ungrounded regions, solving the accuracy-completeness trade-off, and leading to joint improvements in fidelity and perceptual quality.
>
> ### Summary
>
> In summary, our work is not about adding segmentation as a prior, but about **re-thinking how spatial priors can be designed and integrated to solve specific, under-addressed issues in text-conditioned super-resolution**. These deliberate design choices (guided by empirical discovery of new gaps) underpin the novelty and effectiveness of our SRSR framework. While our modules are deliberately simple and efficient, we believe their targeted, insight-driven design is what makes them both practical and genuinely novel.
>
> ## W2: Length of Writing
>
> We appreciate the reviewer’s suggestion to further improve the writing by ensuring that each section is appropriately proportioned. We are committed to making the paper as accessible and polished as possible. In response to this helpful feedback, we will revise the abstract to provide a more complete overview and streamline the first paragraph of the introduction to improve flow and readability.
>
> > “Existing Stable-Diffusion-based super-resolution approaches often exhibit semantic ambiguities due to inaccuracies and incompleteness in their text prompts, coupled with the inherent tendency for cross-attention to divert towards irrelevant pixels. **These limitations can lead to semantic misalignment and hallucinated details in the generated high-resolution outputs.** To address these, we propose a novel, plug-and-play *spatially re-focused super-resolution (SRSR)* framework **that consists of two core components: first, we introduce *Spatially Re-focused Cross-Attention (SRCA)***, which refines text conditioning at inference time by applying visually-grounded segmentation masks to guide cross-attention. **Second, we introduce a *Spatially Targeted Classifier-Free Guidance (STCFG)* mechanism that** selectively bypasses text influences on ungrounded pixels to prevent hallucinations. Extensive experiments on both synthetic and real-world datasets demonstrate that SRSR consistently outperforms seven state-of-the-art baselines in standard fidelity metrics (PSNR and SSIM) across all datasets, and in perceptual quality measures (LPIPS and DISTS) on two real-world benchmarks, underscoring its effectiveness in achieving both high semantic fidelity and perceptual quality in super-resolution.”
> >
>
> We believe this revised abstract better balances clarity and completeness, and remain open to further improvements based on feedback.
>
> ## W3: Explanation of Figure 1
>
> We thank the reviewer for this observation. The regions not highlighted in the attention matrix correspond to areas that could not be confidently grounded by Grounded SAM using the input tag (in this case, “bird”). Since Grounded SAM operates directly on the low-resolution (LR) input, degradation in the image can cause it to be more conservative in assigning segmentation masks, resulting in certain relevant regions (such as the bird’s body) being left ungrounded.
>
> We refer to such areas as **ungrounded regions** throughout the paper. We wish to highlight and reassure the reviewer that we have already analyzed this limitation in **Section 3.3** and specifically designed STCFG to bypass text influence in ungrounded regions, preventing hallucinations and reducing the negative impact of incomplete coverage.
>
> ## W4: Additional References
>
> We thank the reviewer for noting these related works, and will discuss both in the revision:
>
> - **SFT** demonstrates how spatially-varying categorical priors can be integrated into SR networks to recover more realistic textures.
> - **CoSeR** highlights the potential of bridging image and language understanding using diffusion-based priors and cognitive embeddings.
>
> While these methods use semantic or language guidance, our SRSR framework introduces a **grounded and spatially-refocused mechanism (SRCA + STCFG)** that directly tackles the challenge of semantic ambiguities caused by diverted cross-attention, inaccurate text prompts, and incomplete/ungrounded text prompts, which are not explicitly addressed by SFT or CoSeR.
>
> ## W5: Why not use open-vocabulary panoramic segmentation models?
>
> We appreciate this suggestion. In fact, we independently explored this exact idea and incorporated it into our work. Specifically, in Sec. 3.3, Sec. 4.3, and Supplementary Sec. B (Figures S12 and S13), we quantitatively and qualitatively evaluated Prompt-Free Anything Detection and Segmentation from DINO-X, a state-of-the-art open-vocabulary segmentation method.
>
> While this approach improves coverage over Mask2Former, its lack of degradation-awareness compromises accuracy on degraded LR inputs, leading to incorrect prompt-region associations and inferior SR performance (see Tab. 2, V8 vs. V4). Supplementary Figs. S12–S13 further illustrate these issues, which our method overcomes. We will clarify this in the revision.
>
> ## W6-W8: Minor writing and formatting issues
>
> Thank you for these helpful observations. We will address all the mentioned concerns in the revision:
>
> - **W6**: We will revise the methodology sub-sections accordingly by highlighting the strength of our proposed SRCA and STCFG modules before the detailed explanations.
> - **W7**: We will revise the tense in line 179 to maintain consistency.
> - **W8**: We acknowledge the duplicated references (9 and 10, 45 and 46) and will remove the redundancies in the updated bibliography.

---

> > ### Comment · Reviewer_pH61 · 2025-08-05
> >
> > Authors provide very nice and detailed rebuttal about novelty and design, which help me understand the work a lot. I will raise my score.

---

> > > ### Author Response · Authors · 2025-08-06
> > >
> > > Thank you very much for your careful reassessment of our submission after the rebuttal. We greatly appreciate your acknowledgment that our clarifications about the novelty and design have significantly improved your understanding of the work and addressed your initial concerns. We're particularly grateful for your willingness to reconsider and raise your evaluation score accordingly.
> > >
> > > Thank you once again for your thoughtful and constructive feedback, which has helped us clarify our key contributions and enhance the overall quality of this work. Please let us know if any further clarifications are needed; we’d be happy to provide them.

---

### Official Review · Reviewer_RFp9 · 2025-07-01

**Clarity:** 3
**Significance:** 3
**Originality:** 3
**Rating:** 5
**Confidence:** 4

**Summary:**

This paper presents SRSR, a plug-and-play and training-free framework that can be easily applied to Stable-Diffusion-based SR methods with text conditions. The proposed method enhances semantic fidelity by grounding tags to guide cross-attention and suppress irrelevant text influences on ungrounded regions. SRSR improves both visual quality and alignment with semantics and also alleviates hallucination, outperforming baselines across multiple datasets and full-reference evaluation metrics.

**Questions:**

1. In Table 2 V5, can you explain how you add Ungrounded Tags? Do these ungrounded tags also associate with segmentation maps? Otherwise, how to use these ungrounded tags with SRCA and STCFG?
2. Do you try to apply the proposed method to other recent works like AddSR and TSD-SR?
3. It appears that the proposed method offers greater improvement to the multiple-step method, such as SeeSR, compared to the single-step method like OSEDiff. Have you conducted experiments to evaluate how the diffusion steps affect the results with the proposed method?

**Ethical Concerns:**

["NO or VERY MINOR ethics concerns only"]

**Final Justification:**

The authors have provided clear and thorough responses that answered my question and addressed my concerns. While the reliance on CFG may limit the applicability of SRSR to some extent, the method remains a valuable and effective plug-in for compatible diffusion SR models. I have updated my score to reflect a more positive assessment of the paper.

**Limitations:**

Yes

**Paper Formatting Concerns:**

No major formatting issues.

**Quality:**

3

**Strengths And Weaknesses:**

# Strengths
+ The proposed method is simple, straightforward, and intuitive.
+ The experiment results demonstrate that the proposed method can be easily plugged into existing text-conditioned Stable-Diffusion-based SR methods and improve the SR quality with minimal additional burden.
+ Overall, the writing and figures are clear and enjoyable to read.
# Weaknesses
+ This paper argues that previous methods using a segmentation model like Mask2Former and Prompt-Free Anything Detection and Segmentation in DINO-X lack degradation-awareness and thus lead to inferior results. However, the Grounded SAM they used is not degradation-aware, either.
+ Grounding shows only slight improvements compared to the baseline, as seen in V1 and V2 of Table 2. This could weaken the argument that the accuracy-completeness trade-off is important and that accuracy typically outweighs completeness.

---

> ### Author Rebuttal · Authors · 2025-07-31
>
> We sincerely thank the reviewer for their thoughtful evaluation and positive assessment of our work regarding several important aspects:
>
> - **Comprehensive experiments and effective results**: “SRSR improves both visual quality and alignment with semantics and also alleviates hallucination, outperforming baselines across multiple datasets and full-reference evaluation metrics,”
> - **Practical merits**: “The experiment results demonstrate that the proposed method can be easily plugged into existing text-conditioned Stable-Diffusion-based SR methods and improve the SR quality with minimal additional burden,” and “a plug-and-play and training-free framework that can be easily applied to Stable-Diffusion-based SR methods with text conditions.”
> - **Intuitive and original methodology**: Recognized by the numerical rating (“Originality: 3: good”) and the encouraging comments: “The proposed method is simple, straightforward, and intuitive,” and “the proposed method **enhances semantic fidelity** by grounding tags to guide cross-attention and suppress irrelevant text influences on ungrounded regions.”
> - **Clarity and presentation**: “Overall, the writing and figures are clear and enjoyable to read.”
>
> Finally, we appreciate the reviewer’s constructive feedback in the weaknesses and questions sections, and we have addressed each point thoroughly in the responses below.
>
> ## W1: Why is our framework degradation-aware?
>
> Thanks for this important concern regarding degradation-awareness. We would like to clarify below the fundamental difference between our proposed framework and the alternative semantic segmentation-based designs (Mask2Former and Prompt-Free Anything Detection and Segmentation in DINO-X), highlighting the key reasons for our superior results.
>
> While we acknowledge that Grounded SAM itself is not inherently degradation-aware, **our framework introduces degradation-awareness explicitly at the prompt extraction stage (prior to employing Grounded SAM)**. Specifically, we utilize a Degradation-Aware Prompt Extractor (DAPE), which is carefully fine-tuned from the Recognize Anything Model (RAM) to ensure extracted text tags are robust and accurate even under degraded, low-resolution (LR) conditions. Consequently, when these degradation-aware tags from DAPE are provided as inputs to Grounded SAM, the resulting prompt–mask pairs inherently exhibit enhanced degradation-awareness and accuracy. In contrast, semantic segmentation methods directly assign segmentation labels as prompts without a separate degradation-aware prompt extraction step, thus limiting their robustness and accuracy for degraded LR images.
>
> To substantiate this claim, we also conducted comprehensive analyses:
>
> - **Quantitative ablation studies (Sec. 4.3, Table 2):** Variants V7 (Mask2Former) and V8 (DINO-X), both lacking degradation-awareness, show clearly inferior results compared to our proposed approach (V4).
> - **Qualitative analysis (Supplementary Sec. B, Figures S12–S15):** For example, Figures S12 and S13 demonstrate how the lack of degradation-awareness in semantic segmentation methods *causes incorrect yet confidently-extracted prompts*, subsequently misguiding restoration and producing semantically incorrect outputs. In contrast, our proposed DAPE + Grounded SAM framework effectively addresses this limitation, yielding substantially improved semantic correctness.
>
> We will ensure these clarifications are further emphasized in the revised version.
>
> ## W2: Grounding and Accuracy-Completeness Trade-off
>
> We appreciate the reviewer’s thoughtful comment regarding grounding and the accuracy-completeness trade-off. Below, we unpack this issue into three parts to comprehensively address your concern:
>
> 1. The importance of the accuracy-completeness trade-off;
> 2. Why accuracy typically outweighs completeness;
> 3. The broader contribution of grounding beyond direct performance improvement.
>
> ### (1) Importance of accuracy-completeness trade-off:
>
> Accuracy and completeness of the extracted prompts are two of the three key research gaps we explicitly identified in Section 3.1. In our preliminary exploration (Table 2, comparing V1 to V2), we found grounding to be a promising direction as it can provide slight improvements to the baseline by simply applying grounding to improve on accuracy at the cost of completeness (without integrating deliberately designed modules such as SRCA and STCFG). At this stage, we also observed that grounding alone yields only small improvements, as the accuracy gains come at the cost of completeness, resulting in modest net gains. This clearly highlighted the necessity of explicitly addressing the accuracy-completeness trade-off. This motivated our design of the STCFG module, which significantly improved this trade-off and overall restoration.
>
> ### (2) Why accuracy typically outweighs completeness:
>
> The importance of accuracy over completeness is much more significant after employing STCFG to address the aforementioned important challenge of the accuracy-completeness trade-off. Our core insight is that reduced completeness mainly affects ungrounded regions. STCFG targets these specifically, reducing issues from incomplete coverage and **making the completeness a much less concerning issue than incorrectness**.
>
> After integrating STCFG (V6), we observed substantial gains compared to V1, showing that optimized accuracy outweighs losses in completeness. Further evidence supporting this claim comes from comparing our DAPE + Grounded SAM approach (V4) against segmentation-based methods (V7 and V8). Although segmentation methods inherently achieve higher completeness, their less accurate (due to poorer degradation-awareness) prompts directly led to inferior overall performance relative to our method.
>
> ### (3) Contribution of grounding beyond direct performance improvement:
>
> Finally, we emphasize that grounding's contribution extends significantly beyond the performance improvements from V1 to V2. Crucially, grounding provides the essential tag–mask pairs necessary for enabling our SRCA and STCFG modules, without which these key innovations would not be feasible.
>
> ## Q1: Implementational detail
>
> We appreciate the opportunity to clarify the implementation of the ablated version V5. V5 first retains the entire settings of V4 and builds on it by additionally including ungrounded tags to demonstrate their negative impact. Specifically, V5 retains the SRCA refinement for grounded regions, where tag–mask pairs are available, and continues to use STCFG to block text influence in ungrounded regions. The only change is that, for V5, in addition to V4, we also keep the ungrounded tags and allow their default (inherent) cross-attention maps, **without applying SRCA**, since these tags do not have associated segmentation maps (as the reviewer correctly noted).
>
> Because STCFG continues to restrict text influence in the ungrounded regions, the effect of these ungrounded tags is limited to their flawed cross-attention behavior within the union of all grounded regions. This setup highlights the drawback of including ungrounded tags: without the benefit of SRCA, their attention may be diffused or misdirected across the object regions, as seen for the “camouflage” tag in the bottom example of Fig. 2 that caused the baseline’s hallucinated object (highlighted in red), where the SRCA map would otherwise be entirely black and help effectively avoid the hallucination.
>
> ## Q2: Application to other works
>
> Thank you for this insightful question. We have not yet applied SRSR to AddSR or TSD-SR, but due to the plug-and-play nature of our framework, SRSR can be flexibly integrated into these and other recent approaches to further enhance their performance. In this work, we demonstrated the generality and effectiveness of SRSR by incorporating it into both SeeSR and OSEDiff, and we observed consistent improvements across multiple benchmarks and evaluation metrics. We will include a discussion in the revised manuscript (Future Work section) to highlight the potential for SRSR to benefit existing methods like AddSR, TSD-SR, and future methods. We believe the community will benefit from broader adoption of SRSR, and we encourage future research to explore its integration with a wider range of Real-ISR frameworks.
>
> ## Q3: Different diffusion steps
>
> Thank you for this valuable question. We would like to clarify that the greater benefit observed for SeeSR, compared to OSEDiff, primarily results from the use of STCFG rather than differences in the number of diffusion steps. Specifically, in OSEDiff, CFG is disabled by default, which renders STCFG inapplicable, meaning the performance gain comes solely from SRCA. In contrast, SeeSR benefits from both SRCA and STCFG, leading to more substantial improvements.
>
> Motivated by the reviewer’s insight, we conducted additional controlled experiments to further evaluate the effect of diffusion steps on performance, keeping all other factors fixed. For both our method and the baseline, we compared results at 50, 40, and 30 sampling steps. As shown in the table below, while reducing the number of diffusion steps introduces the typical fidelity–perceptual quality trade-off (i.e., fewer steps improve on fidelity metrics PSNR/SSIM but worsen perceptual quality metrics LPIPS/DISTS for both methods), our method consistently maintains a significant and stable advantage over the baseline across all metrics and step choices.
>
> | Version | PSNR↑ | SSIM↑ | LPIPS↓ | DISTS↓ |
> | --- | --- | --- | --- | --- |
> | Ours-s50 | **26.3996** | **0.7632** | **0.2718** | **0.2092** |
> | SeeSR-s50 | 25.1717 | 0.7219 | 0.3008 | 0.2223 |
> |  |  |  |  |  |
> | Ours-s40 | **26.5189** | **0.7644** | **0.2751** | **0.2122** |
> | SeeSR-s40 | 25.3804 | 0.7290 | 0.3027 | 0.2225 |
> |  |  |  |  |  |
> | Ours-s30 | **26.7346** | **0.7670** | **0.2777** | **0.2164** |
> | SeeSR-s30 | 25.5453 | 0.7316 | 0.3059 | 0.2272 |
> |  |  |  |  |  |

---

> > ### Comment · Reviewer_RFp9 · 2025-08-05
> >
> > Thank you for your thoughtful and detailed response. Your clarifications have greatly improved my understanding of the proposed method and have addressed many of my initial concerns.
> >
> > For Q3, as a follow-up, I’m curious why OSEDiff disables CFG by default. Is CFG not compatible with OSEDiff? Or perhaps adding CFG has a negative impact on OSEDiff?

---

> > > ### Author Response · Authors · 2025-08-05
> > >
> > > Thank you for your thoughtful review of our rebuttal and for letting us know that our rebuttal effectively addressed your initial concerns.
> > >
> > > Regarding your follow-up question on CFG in OSEDiff, your first suggestion is correct: CFG is indeed incompatible with OSEDiff. Specifically, OSEDiff was not trained in a manner that supports CFG, rather than intentionally disabling it to avoid negative effects (as your second suggestion implied).
> > >
> > > Thank you again for your valuable feedback. Please let us know if there are any further questions, and we'd be happy to address them.

---

> > > > ### Comment · Reviewer_RFp9 · 2025-08-05
> > > >
> > > > Thank you for your clarifications. Is it fair to say that, although SRSR is not strictly limited to models using CFG, its full potential is best realized when the underlying model supports CFG? In cases where CFG is not available, such as with OSEDiff, the effectiveness of SRSR appears to be reduced, as STCFG cannot be applied. This is particularly notable given that STCFG is the primary contributor to the performance gains based on the ablation study in Table 2.

---

> > > > > ### Author Response · Authors · 2025-08-06
> > > > >
> > > > > Thank you for this insightful follow-up. You are absolutely correct—indeed, SRSR is not strictly limited to models using CFG, and the support of CFG (and thus our STCFG module) is precisely why the SeeSR baseline realizes the full potential of SRSR and achieves greater performance gains compared to OSEDiff, rather than differences in diffusion steps as initially assumed in Q3.
> > > > >
> > > > > We also wish to highlight that even in cases where CFG is unavailable, our proposed SRCA module alone still consistently enhances the fidelity of the restored images, demonstrating the versatility and general applicability of SRSR.

---

> > > > > > ### Comment · Reviewer_RFp9 · 2025-08-06
> > > > > >
> > > > > > Thank you for your detailed response. I agree that even when CFG is not compatible, SRSR remains a useful plug-in module, as SRCA can still provide performance improvement. However, the gains from SRCA alone are noticeably smaller than those achieved with STCFG. Therefore, I believe it would be helpful to highlight in the paper that, while SRSR can be applied to any cross-attention-based diffusion SR approach utilizing text priors, the full potential of the method is only realized when CFG is available.
> > > > > >
> > > > > > Overall, the authors have addressed my concerns and questions thoroughly. I appreciate their clarifications, and I will raise my score accordingly.

---

> > > > > > > ### Author Response · Authors · 2025-08-06
> > > > > > >
> > > > > > > Thank you very much for this valuable suggestion and for all your thoughtful effort in reviewing our submission. We fully agree and will explicitly highlight this in our revision, as you kindly suggested. We believe emphasizing this point will improve the clarity and practical guidance of our paper. We are also delighted to hear that we have effectively addressed all your concerns and questions. Thank you once again for raising your final rating accordingly, and for your consistently positive evaluation of this work.

---

### Comment · Area_Chair_mc1v · 2025-08-04
**Please Read the Rebuttal and Discuss**

Dear Reviewers RFp9, pFBn. pH61, u37U

The authors have submitted their rebuttal.

Please carefully review all other reviews and the authors’ responses, and engage in an open exchange with the authors.

Kindly post your initial response as early as possible within the discussion window to allow sufficient time for interaction.

Your AC

---

### Decision · Program_Chairs · 2025-09-17

**Decision:**

Accept (poster)

**Comment:**

The paper proposes SRSR, a plug-and-play framework that enhances semantic accuracy in text-conditioned Real-ISR through two key modules (SRCA and STCFG). Reviewers initially raised concerns about novelty, reliance on previous works DAPE and Grounded SAM, and the considerable drop in no-reference metrics. The rebuttal clarified the methodological contributions, degradation-awareness design, and the perception–fidelity trade-off, with additional ablation and metric analyses. All reviewers acknowledged the clarifications, raised their scores, and now recommend acceptance. Overall, this is a solid and practically useful paper with strong empirical results, and valuable contributions to the Real-ISR community.